# Distinct ventral tegmental area neuronal ensembles are indispensable for reward-driven approach and stress-driven avoidance behaviors

Ioannis Koutlas ⊙, Lefkothea Patrikiou ⊙, Stef E. van der Starre, Diaz Danko, Inge G. Wolterink-Donselaar, Mieneke C. M. Luijendijk ⊙, Roger A. H. Adan ⊙ & Frank J. Meye ⊙ ✉

Assigning valence to stimuli for adaptive behavior is an essential function, involving the ventral tegmental area (VTA). VTA cell types are often defined through neurotransmitters (NT). However, valence function in VTA does not parse along NT-boundaries as, within each NT-class, certain neurons are excited by reward and others by stressors. Here we identify, in male mice, the co-activated VTA neuronal ensembles for reward and stress, and determine their role in adaptive behaviors. We show that stimuli of opposite valence (opioid vs acute social stress) recruit two distinct VTA neuronal ensembles. These two ensembles continue to be preferentially engaged by congruent valence stimuli. Stimulation of VTA stress- or reward ensembles is aversive/ reinforcing, respectively. Strikingly, external valence stimuli fully require activity of these small discrete VTA ensembles for conferring approach/ avoidance outcomes. Overall, our study identifies distinct VTA ensembles for positive and negative valence coding and shows their indispensability for adaptive behavior.

Attributing valence to rewarding and aversive stimuli is essential for adaptive behavior, and this process goes awry in psychiatric diseases, such as depression, addiction, and eating disorders[1–4]. The ventral tegmental area (VTA) is implicated in these disorders, and plays an important role in processing the valence of stimuli[5–8]. Much research has focused on which cell types in the VTA respond to rewarding and which to aversive stimuli. Often such cell type classification is done primarily based on neurotransmitter (NT) content and projection targets. However, it is becoming increasingly clear that the functions of coding reward and of aversion do not easily segregate along these lines in the VTA, and are guided by more nuanced principles.

The VTA is a heterogeneous region comprising distinct neuronal subpopulations characterized by dopamine (DA), gamma-aminobutyric acid (GABA), and glutamate (GLU) neurotransmitter profiles, as well as combinatorial types[9]. Traditionally, $VTA_{DA}$ neurons have been shown to be excited by unexpected rewards[10–12]. These cells typically respond to aversive stimuli with a decrease in activity[7,10,11]. However, there are also $VTA_{DA}$ neurons which are excited by aversive stimuli[13–17]. There is also evidence that some $VTA_{GLU}$ neurons respond to rewarding stimuli and others to aversive stimuli[12,18,19]. Also, $VTA_{GABA}$ neurons can be excited either by rewarding[7] or aversive stimuli[7,20].

Therefore, a reward or stress experience will co-activate a set of VTA neurons with cellular identities that transcend NT-categories (i.e., ensembles[21]). Accordingly, previous work showed that morphine and foot shock engage multiple neuronal cell types in VTA[22], and we and others previously showed that naturalistic social stress engages an NT-transcending ensemble in VTA[20,23]. An important remaining challenge in the field is to show the extent to which these reward- and stress-

Department of Translational Neuroscience, Brain Center, UMC Utrecht, Utrecht University, Utrecht, the Netherlands. ✉e-mail: f.j.meye-2@umcutrecht.nl

responsive VTA ensembles are important in orchestrating adaptive behavioral outcomes.

To address these issues our study focused on specific VTA neuronal ensembles activated by acute systemic administration of the opioid DAMGO (a stimulus of positive valence) or an exposure to acute social stress (a stimulus of negative valence). We utilized Fos-CreER[T2] transgenic (TRAP2) mice[24], whereby c-Fos promoter activity triggers expression of (4OH)-Tamoxifen-dependent Cre recombinase, allowing for experience-dependent neuronal tagging. With immunohistochemical analysis we show that the small ensembles (~10% of all VTA neurons) activated by social stress (VTA$_{STRESS}$) or by systemic DAMGO (VTA$_{DAMGO}$) are topographically intermingled, have similarly diverse NT-compositions, but nonetheless represent predominantly non-overlapping populations. Fiber photometric calcium recordings from these ensembles show them to have divergent responses to food reward and foot shocks. Chemo-/optogenetic reactivation of these ensembles is sufficient to induce behavioral approach/avoidance responses that mimic the integral experience of DAMGO or social stress, respectively. Crucially, the opposite behavioral effects on approach and avoidance elicited by DAMGO and social stress, were completely blocked by chemogenetic inhibition of just the small VTA$_{DAMGO}$ or VTA$_{STRESS}$ ensemble, respectively.

Overall, these findings further elucidate the ensemble architecture of the ventral tegmental area for valence coding, and show the remarkable potency and indispensability of these specific neuronal sets in orchestrating adaptive behavioral repertoires.

## Results

### Stimuli of opposite valence differentially affect approach/avoidance behavioral domains

To determine the neuronal ensembles in the VTA that respond to a rewarding or an aversive stimulus, we first established two contrasting experiences. We tested whether a single systemic injection of DAMGO, a selective μ-opioid receptor agonist, or an acute social stressor would trigger contrasting effects on behavioral tests for valence, anxiety and feeding.

We first tested C57BL/6 J mice using a single exposure place preference (or avoidance) paradigm (sePP/sePA), as described before[25,26]. For DAMGO-conditioning, mice were randomly placed in one of two compartments on day 1, after either an i.p. DAMGO (or vehicle) injection. On day 2, the mice received vehicle (or DAMGO) in the other compartment. On day 3 we assessed their preference score for the compartments when they were free to explore (Fig. 1A). Mice spent significantly more time in the compartment previously paired with a DAMGO injection (Fig. 1B). For social stress-conditioning, mice were exposed to either an aggressive CD-1 mouse in a compartment, or were in a compartment without another mouse. Also these experiences were randomly counterbalanced across day 1 and 2. We observed that on day 3, these mice avoided the compartment formerly paired with this social stress experience (Fig. 1C).

We then assessed whether DAMGO and acute social stress would elicit opposite effects on anxiety-like behavior in an open field (OF) test (Fig. 1D). Mice receiving DAMGO prior to the OF, as compared to vehicle, spent more time in the center of the arena, suggesting that systemic DAMGO administration induced an anxiolytic phenotype (Fig. 1E, F; Suppl. Figure 1A). This occurred in the absence of general locomotor effects in the arena (Suppl. Figure 1B, C). Conversely, mice that were acutely stressed spent less time in the center of the OF compared to their respective controls (Fig. 1E, G; Suppl. Figure 1D), indicating that acute social stress induced an anxiogenic phenotype. This was accompanied by a stress-induced reduction in general locomotion along with heightened immobility, as shown before[27] (Suppl. Fig. 1E, F).

Finally, we examined whether these experiences acutely alter the intake of palatable food (Fig. 1H). Animals were administered DAMGO or were exposed to social stress, and were subsequently placed in a novel cage with both their regular chow pellet diet and a palatable food reward (fat). We then monitored their food choices and intake. Neither DAMGO nor social stress altered chow intake (Suppl. Fig. 1G, H). Instead, we found that DAMGO increased fat intake, compared to baseline and to a saline injection (Fig. 1I). Conversely, mice that were acutely stressed consumed significantly less fat in the first hour post-stress, compared to their baseline intake levels and to mice in the control condition (Fig. 1J).

Overall, these data suggest that administration of DAMGO and acute exposure to social stress are experiences with opposite effects on valence, anxiety levels, and food intake.

### Stimuli of opposite valence engage distinct VTA ensembles

Having established that systemic DAMGO and acute social stress represent opposite experiences for the mice, we then asked whether these stimuli would recruit distinct neuronal ensembles within the VTA. To this end, we injected mice with DAMGO or exposed them to an acute social stress episode and subsequently perfused them (Fig. 2A). We then did immunohistochemical staining of VTA slices against Fos to gauge neuronal activation, and against tyrosine hydroxylase (TH) to demarcate the boundaries of the VTA (Fig. 2B). We previously showed that stress elevates Fos levels in the VTA[23]. Here we observe that DAMGO led to similar amounts of Fos positive neurons in VTA as did social stress (Fig. 2C). We from here on refer to these collectives of cells as the VTA$_{DAMGO}$ and VTA$_{STRESS}$ ensembles, respectively.

We then characterized the cell types within these VTA$_{DAMGO}$ and VTA$_{STRESS}$ ensembles. Similarly to our previous findings on social stress ensembles[23], the VTA$_{DAMGO}$ ensemble was found to be predominantly composed of neuronal cells (~98% NeuN positive; Suppl. Fig. 2A, B). The size of this ensemble was ~10% of all VTA neurons (Suppl. Figure 2C), similar to what we previously reported for the VTA$_{STRESS}$ ensemble (~11%)[23]. Having established that both ensembles are predominantly neuronal, we then assessed the neurotransmitter content within them. To identify VTA$_{DA}$ neurons we performed immunohistochemistry against TH in C57BL/6 J mice. To identify VTA$_{GABA}$ and VTA$_{GLU}$ neurons we used vesicular GABA transporter (vgat)-Cre and vesicular glutamate 2 transporter (vglut2)-Cre transgenic mouse lines, respectively. We injected these in the VTA with an Adeno Associated Virus (AAV) to Cre-dependently drive expression of a fluorophore (AAV-DIO-mCherry) (Fig. 2D, Suppl. Fig. 2D–E).

Both ensembles contained the same extent of dopaminergic (~50%) and GABAergic (<12%) neurons (Fig. 2E). Both ensembles also contained a similar extent of combinatorial glutamatergic/dopaminergic neurons (<12%). The only difference between the ensembles was that the VTA$_{DAMGO}$ ensemble had a slightly higher degree of glutamatergic/TH-negative cells (17% VTA$_{DAMGO}$ vs 7% in VTA$_{STRESS}$) (Fig. 2E). We next analyzed the topographical position of VTA$_{DAMGO}$ and VTA$_{STRESS}$ cells. Such analysis, across the rostral-caudal axis, revealed no significant differences in the average position of DAMGO- and stress-ensemble cells in the medio-lateral and dorso-ventral axis (Suppl. Figures 2F–H). Overall, the ensembles therefore contained largely similar NT-defined cell types and had a similar anatomical topography.

Next, we evaluated whether these two ensembles, which on multiple levels of analysis appear similar, were actually distinct from each other (or instead were the same overlapping ensemble). To address this, we sought to determine the extent to which an ensemble for stress would show reactivity to reward, and vice versa. For this we used Targeted Recombination in Active Populations (TRAP2)[24], a technique that employs tamoxifen-inducible Cre recombinase under the control of an immediate early gene promoter to label neurons activated during specific experiences. By combining TRAP2 with subsequent Fos immunohistochemistry, we aimed to capture and compare neuronal ensembles activated by two distinct experiences at different time points (Fig. 2F).

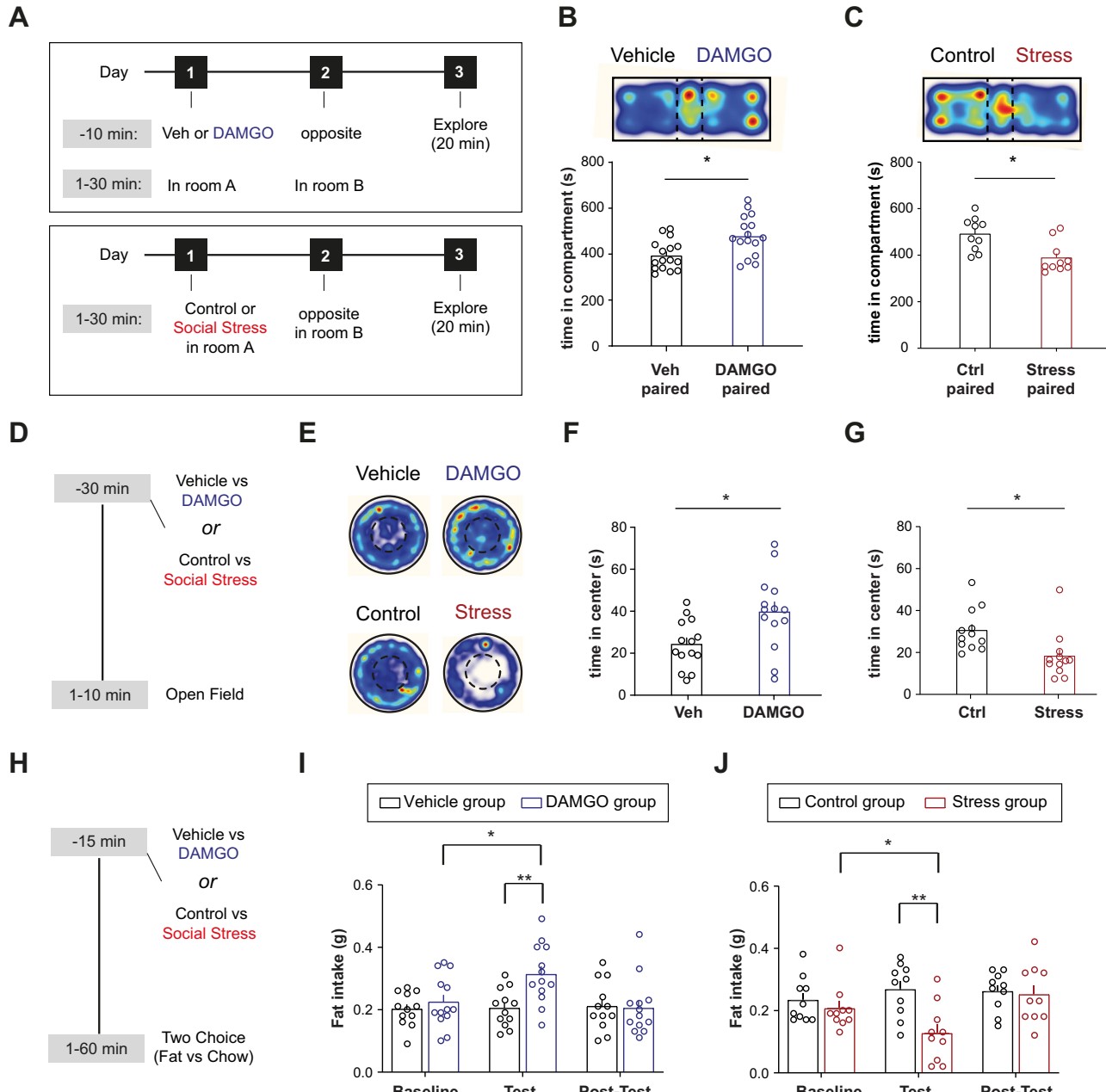

**Fig. 1 | Stimuli of opposite valence differentially affect approach/avoidance behavioral domains. A** Single episode place preference/avoidance (sePP/sePA) experimental timeline for vehicle vs DAMGO and control vs social stress experiments. **B** DAMGO-induced sePP. Top: Representative heat map of place preference arena from a mouse that received DAMGO in the right compartment. Bottom: Average time spent in each arena compartment (N = 16, paired t test, t(15) = 2.305, p = 0.0359). **C** Stress-induced sePA. Top: Representative heat map of place preference arena from a mouse exposed to a CD1 aggressor in the right compartment. Bottom: Bar graph showing the average time spent in each arena compartment (N = 10, paired t test, t(9) = 2.478, p = 0.0351). **D** Open field test experimental timeline. **E** Top: Representative heat maps from mice that received vehicle or DAMGO prior to testing. Bottom: Representative heat maps from mice exposed to a novel male C57BL/6 J conspecific or social stress prior to testing. **F** Time spent in the center of the open field after vehicle or DAMGO injection

(N$_{veh}$=14, N$_{DAMGO}$ = 14, unpaired t test, t(26) = 2.701, p = 0.012). **G** Time spent in the center of the open field after control or stress exposure (N$_{ctrl}$=12, N$_{Stress}$ = 12, unpaired t test, t(22) = 2.812, p = 0.0101. This effect persisted when distance moved was taken along as a covariate, F(1,21) = 7.089, p = 0.015). **H** Fat intake experimental timeline. **I** Amount of fat consumed during the 1h-long binge session for baseline, test and post-test days (N$_{veh}$=12, N$_{DAMGO}$ = 13, Two-way ANOVA, Day x Ligand interaction, F(2,46) = 5.01, p = 0.0108, Tukey's post-hoc for DAMGO: baseline vs test, q(12) = 4.159, p = 0.0308). Mice in the DAMGO group received DAMGO only during test day. **J** Amount of fat consumed during the 1h-long binge session for baseline, test and post-test days (N$_{ctrl}$=10, N$_{Stress}$ = 10, Two-way ANOVA, Day x Group interaction, F(2,36) = 7.071, p = 0.0026, Tukey's post-hoc for Stress: baseline vs test, q(36) = 4.223, p = 0.0137). Data are presented as mean values + SEM. All statistical tests were performed two-sided. Source data are provided as a Source Data file.

First, we showed that both DAMGO and acute social stress, followed by an injection of 4-hydroxytamoxifen (4-OHT), led to tagging of VTA neuronal ensembles (Fig. 2G). The number of TRAPed cells for either DAMGO or social stress was approximately one third of that of the endogenous Fos-expressing neurons (Suppl. Fig. 2I, J; see Fig. 2C), as we saw before for the social stress experience[23]. Notably though, the TRAP-captured ensembles for both experiences were still of similar size (Suppl. Figure 2K). Furthermore, similarly to endogenous Fos

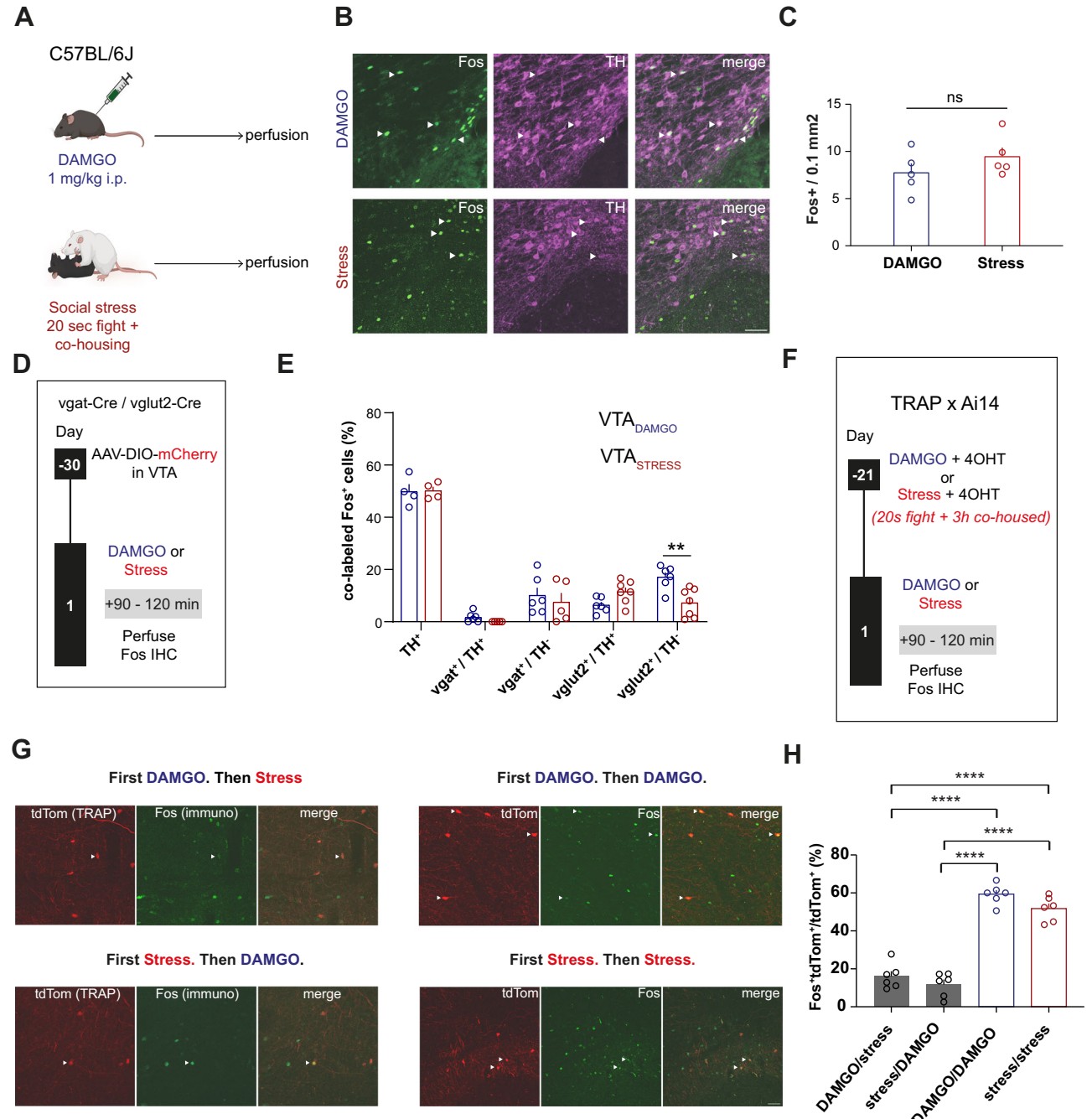

**Fig. 2 | Stimuli of opposite valence engage distinct VTA ensembles.**
**A** Experimental timeline for DAMGO- and social stress-driven Fos induction in the VTA. **B** Representative examples of immunohistochemical staining against Fos and tyrosine hydroxylase (TH) in the VTA for DAMGO (top) and Stress (bottom) groups. Scale bar: 50 μm. **C** Number of Fos$^+$ cells per 0.1 mm$^2$ in the VTA (N = 5 per group, unpaired t test, t(8) = 1.248, p = 0.2474). **D** Timeline of experiments to fluorescently label GABAergic (vgat$^+$) and glutamatergic (vglut2$^+$) neurons in the VTA and identify Fos$^+$ subsets after DAMGO or social stress **E** Cell type composition of VTA$_{DAMGO}$ and VTA$_{STRESS}$ ensemble. Proportion of TH$^+$, vgat$^+$ with or without TH and vglut2$^+$ with or without TH within the VTA$_{DAMGO}$ and VTA$_{STRESS}$ ensembles (for TH: N$_{DAMGO}$ = 4, N$_{STRESS}$ = 4, for Vgat/TH: N$_{DAMGO}$ = 6, N$_{STRESS}$ = 5, for Vglut2/TH: N$_{DAMGO}$ = 6, N$_{STRESS}$ = 7; Two-way ANOVA, F(4,46) = 3.934, p = 0.0079, Sidak's post-hoc contrast

for vglut2$^+$/TH$^-$ (DAMGO vs Stress), t(46) = 3.625, p = 0.0036. **F** Experimental timeline for tagging distinct ensembles. **G** Representative images of VTA slices from TRAP2xAi14 animals expressing tdTomato in DAMGO- or stress-activated neurons (TRAP) and Fos in DAMGO- or stress-activated neurons during a second exposure. White arrowheads indicate co-localization of tdTomato (TRAP) and Fos. Scale bar: 50 μm. **H** Percentage of double tdTom$^+$ and Fos$^+$ / tdTom$^+$ cells (N = 6 per group, One-way ANOVA, F(3,20) = 92.42, p < 0.0001, Tukey's post-hoc comparisons: DAMGO/stress vs DAMGO/DAMGO, q(20) = 17.12, p < 0.0001, DAMGO/stress vs stress/stress, q(20) = 14.12, p < 0.0001, stress/DAMGO vs DAMGO/DAMGO, q(20) = 18.82, p < 0.0001, stress/DAMGO vs stress/stress, q(20) = 15.82, p < 0.0001). Data are presented as mean values + SEM. All statistical tests were performed two-sided. Source data are provided as a Source Data file.

expression (Fig. 2E), approximately half of each TRAPed ensemble was dopaminergic (Suppl. Figure 2L). We evaluated, >3 weeks later, for these two ensembles how they responded in terms of Fos expression to a repetition of the same stimulus, or instead to the other stimulus.

We observed that the VTA$_{DAMGO}$ ensemble was more reactive to a subsequent DAMGO exposure than to subsequent stress (Fig. 2G, H, Suppl. Figure 2I). Similarly, the VTA$_{STRESS}$ ensemble was more reactive to subsequent social stress than to subsequent DAMGO (Fig. 2G, H,

Suppl. Figure 2I). Importantly, these differences in preferential activation patterns were not due to any different extents of overall VTA Fos or TRAP densities between the groups (Suppl. Fig. 2J, K).

To further investigate valence-specific encoding in ensembles initially exposed to either DAMGO or social stress (TRAP), we examined their responses to congruent valence experiences beyond re-exposure to the same stimulus. Mice initially exposed to DAMGO received a 2-hour food reward session (fat, after overnight food restriction), while those exposed to social stress experienced an aversive foot shock. We then perfused the mice for Fos staining. VTA ensembles responded preferentially to these congruent valence experiences (food reward for DAMGO-exposed mice; foot shock for social stress-exposed mice) versus incongruent ones (social stress or DAMGO, respectively) (Suppl. Fig. 2M–P). To determine whether the extent of overlap of TRAP and Fos signals for non-congruent valence stimuli (e.g., DAMGO TRAP and social stress Fos or vice versa) was higher than in naïve conditions, we also examined the overlap of these signals in response to home cage exposure. We observed that this also resulted in comparable levels of overlap as to that observed with incongruent valence stimuli, whereas higher overlap was observed with congruent valence stimuli (Suppl. Fig. 2Q; Fig. 2H).

Overall, these data suggest that DAMGO and acute social stress, which produce opposite behavioral phenotypes, recruit two functionally distinct VTA neuronal ensembles, that have similarities in terms of NT-content, neuronal density, and topography.

## $VTA_{DAMGO}$ and $VTA_{STRESS}$ ensembles differentially respond to stimuli of positive and negative valence in vivo

Our reactivation studies based on Fos, an indirect and static proxy of elevated neural activity, suggested preferential recruitment of the two VTA ensembles by stimuli of congruent valence. Next, we sought to extend these findings, by assessing with dynamic measurements whether the $VTA_{DAMGO}$ ensemble would be (preferentially) acutely activated by another reward, and the $VTA_{STRESS}$ ensemble by another aversive stimulus. To examine this we measured the activity of $VTA_{DAMGO}$ or $VTA_{STRESS}$ ensembles during the intake of palatable food or during foot shocks. To this end, TRAP2 mice were stereotactically injected in the VTA with an AAV driving Cre-dependent recombination of a genetically encoded calcium indicator (AAV-FLEX-jGCaMP8s). An optic fiber was also implanted above the VTA (Fig. 3A, B, Suppl. Figure 3A). After a recovery period, mice were exposed to a DAMGO injection or to social stress followed by a 4-OHT injection, to drive expression of jGCaMP8s specifically in $VTA_{DAMGO}$ or $VTA_{STRESS}$ ensembles (Fig. 3A, B).

We first assessed whether the two ensembles are activated by palatable food intake. We placed mice in an apparatus in which there was fat in one of a total of five ports (the other 4 ports were empty), each with infrared beams to detect entry. We then time-locked entry of the mouse in the reward-containing port while measuring GCaMP8s responses of the VTA ensembles. We observed that $VTA_{DAMGO}$ neurons showed calcium transients around interactions with the fat port (Fig. 3C, D, Suppl. Figure 3B). On the other hand, $VTA_{STRESS}$ neurons showed no calcium activity around interaction with the fat (Fig. 3C, D, Suppl. Figure 3C), suggesting that the ensembles differ in their encoding of this rewarding stimulus. Importantly, nose-poking into an empty (no fat) port did not cause a calcium response in the $VTA_{DAMGO}$ ensemble (Suppl. Figure 3D). We then exposed mice to electric foot shocks and recorded GCaMP8s activity around the time of shock onset. The $VTA_{STRESS}$ ensemble was robustly activated by foot shocks (Fig. 3E, F, Suppl. Figure 3E). Although the $VTA_{DAMGO}$ ensemble also showed reactivity to the foot shocks, this fell short of significance, and this response was significantly smaller than the response by the $VTA_{STRESS}$ ensemble (Fig. 3E, F, Suppl. Figure 3E, F).

Overall, our results demonstrate that $VTA_{DAMGO}$ neurons also show activity during interactions with another positive valence stimulus (a naturalistic food reward), while $VTA_{STRESS}$ neurons are also strongly reactive to another negative valence stimulus other than social stress (foot shocks).

## Stimulation of $VTA_{DAMGO}$ and $VTA_{STRESS}$ ensembles drives congruent approach/avoidance behaviors

We next investigated whether specifically stimulating $VTA_{DAMGO}$ and $VTA_{STRESS}$ ensembles could recapitulate the distinct valence/anxiety/feeding phenotypes, that we observed earlier after exposure of the animal to the integral experience of either systemic DAMGO or social stress (see Fig. 1B–J). For that we bilaterally injected TRAP2 mice in the VTA with a Cre-dependent activating chemogenetic vector (AAV-DIO-hM3Dq) or a Cre-dependent control vector (AAV-DIO-mCherry). We then exposed the animals to a DAMGO injection or to acute social stress, followed by a 4-OHT injection, for permanent expression of activating DREADD or control proteins in $VTA_{DAMGO}$ and $VTA_{STRESS}$ ensembles (Fig. 4A, B).

We first gauged the valence associated with activation of each ensemble, using a conditioned place preference/avoidance paradigm (CPP/CPA; Suppl. Figure 4A). Following a pre-test session on day 1, mice were conditioned for 3 days in the CPP apparatus, where DREADD agonist compound 21 (c21) injection was paired with a specific compartment. During test day (day 5), mice were free to roam in both compartments. Mice showed significant preference for the compartment that had been paired with chemogenetic activation of the $VTA_{DAMGO}$ ensemble. Conversely, mice showed significant avoidance of the compartment that had been paired with chemogenetic activation of the $VTA_{STRESS}$ ensemble (Fig. 4C, D). This indicates that activation of the $VTA_{DAMGO}$ ensemble is rewarding, while activation of the $VTA_{STRESS}$ ensemble is aversive. Control mice that only expressed an mCherry fluorophore in the $VTA_{STRESS}$ ensemble showed no significant preference for any compartment, indicating that c21 alone has no effect on valence processing (Fig. 4D). Importantly, these effects could not be explained by difference in viral expression (ensemble size), as no significant correlation was found between hM3Dq expression (ensemble size) and CPP/CPA score (Suppl. Figure 4B).

Next, we assessed whether the activation of these two VTA ensembles would alter general anxiety in the OF test (Fig. 4E; Suppl. Figure 4C). Administration of c21 in control AAV-DIO-mCherry-expressing mice produced no effect on anxiety-like behaviors (Fig. 4F, Suppl. Figure 4D). Notably, $VTA_{DAMGO}$ ensemble activation led to anxiolysis (as shown by increased time spent in the center of the OF arena), whereas activation of the $VTA_{STRESS}$ ensemble resulted in anxiogenesis, as assessed by decreased time in the center of the arena (Fig. 4G, H, Suppl. Fig. 4E, F). Importantly, locomotor parameters (distance moved, immobility) were unaffected and could not explain the differences between groups (Suppl. Figure 4D–F).

Subsequently, we determined the effect of ensemble activation on reward intake in the palatable food choice paradigm. The mice were placed daily in a novel cage for a period of 1 h where they had free access to both their regular chow food and palatable fat reward. Administration of c21 in mCherry-expressing controls produced no effects on food intake (Fig. 4I, Suppl. Figure 4G). Activation of the $VTA_{DAMGO}$ ensemble had no effect on fat intake (Fig. 4J, Suppl. Figure 4H), in contrast to systemic DAMGO-driven hyperphagia (see Fig. 1I). Notably, activation of the $VTA_{STRESS}$ ensemble led to a decrease of fat intake compared to baseline consumption (Fig. 4K, Suppl. Figure 4I), in accordance with the acute stress-driven hypophagia (see Fig. 1J).

To further explore the roles of $VTA_{DAMGO}$ and $VTA_{STRESS}$ ensembles in encoding (aspects of) positive or negative valence, an operant task involving intracranial self-stimulation (ICSS) of each ensemble was employed. TRAP2 mice were injected with AAV-hSyn-CoChR-GFP in the VTA and optic fibers were bilaterally implanted above the VTA (Fig. 4L, M). After a recovery period mice were exposed to DAMGO or

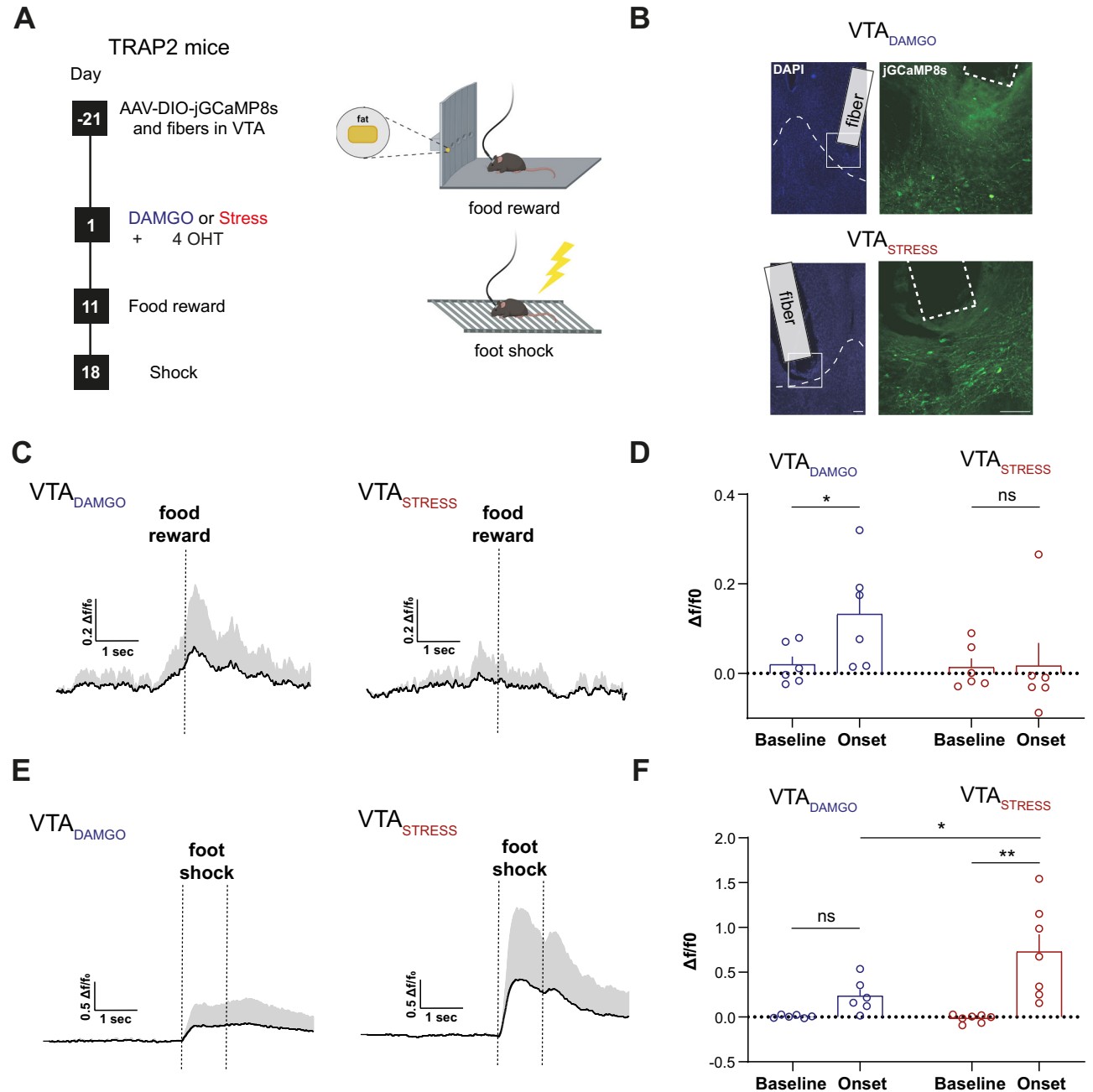

**Fig. 3 | VTA$_{DAMGO}$ and VTA$_{STRESS}$ ensembles differentially respond to stimuli of positive and negative valence in vivo. A** General experimental timeline. **B** Representative examples showing expression of jGCaMP8s in either VTA$_{DAMGO}$ (left) or VTA$_{STRESS}$ (right) ensembles and fiber traces. Scale bar: 100 μm. **C** Average Δf/F$_0$ from VTA$_{DAMGO}$ (left) and VTA$_{STRESS}$ (right) jGCaMP8s-expressing neurons, time-locked around food reward approach. **D** Average Δf/F$_0$ jGCaMP8s response during baseline and food reward interaction onset for VTA$_{DAMGO}$ and VTA$_{STRESS}$ (N = 6 per group, RM Two-way ANOVA, Time Main effect, F(1,10) = 5.02, p = 0.049, Sidak's post-hoc comparisons: baseline vs onset for VTA$_{DAMGO}$, t(10) = 3.078, p = 0.0232; baseline

vs onset for VTA$_{STRESS}$, t(10) = 0.0908, p = 0.995. **E** Average Δf/F$_0$ from VTA$_{DAMGO}$ (left) and VTA$_{STRESS}$ (right) jGCaMP8s-expressing neurons, time-locked at foot shock onset. **F** Average Δf/F$_0$ jGCaMP8s response during baseline foot shock onset for VTA$_{DAMGO}$ and VTA$_{STRESS}$ (N$_{DAMGO}$ = 6, N$_{STRESS}$ = 7, RM Two-way ANOVA, Time x Ensemble interaction, F(1,11) = 4.895, p = 0.049, Sidak's post-hoc comparisons: baseline vs onset for VTA$_{DAMGO}$, t(11) = 1.324, p = 0.3794; baseline vs onset for VTA$_{STRESS}$, t(11) = 4.687, p = 0.0013, VTA$_{DAMGO}$ vs VTA$_{STRESS}$ for onset, t(22) = 3.104, p = 0.0103. Data are presented as mean values + SEM. All statistical tests were performed two-sided. Source data are provided as a Source Data file.

social stress followed by an injection of 4-OHT, to express an opsin in VTA$_{DAMGO}$ or VTA$_{STRESS}$ neurons, respectively. Mice were then placed in an operant cage where nose pokes in an active port of a 5-port wall resulted in ensemble stimulation (473 nm laser). Nose-poking in any of the 4 non-active ports had no programmed consequences. We found that mice with the stimulatory opsin in the VTA$_{DAMGO}$ ensemble nose poked significantly more in the active versus the inactive ports (Fig. 4N top; Suppl. Figure 4J top). To rule out potential confounds linked to

light stimulation, we showed that mice instead did not nose poke when this was paired with laser output at a wavelength that was not effective for opsin stimulation (i.e., 532 nm light delivery) (Suppl. Figure 4K). Furthermore, we found that mice did not seek optogenetic stimulation of the VTA$_{STRESS}$ ensemble (Fig. 4N top, Suppl. Figure 4J top).

In a subsequent experiment we switched the contingencies of stimulation, such that now mice experienced continuous stimulation of the ensemble with the 473 nm laser in the operant chamber. Under

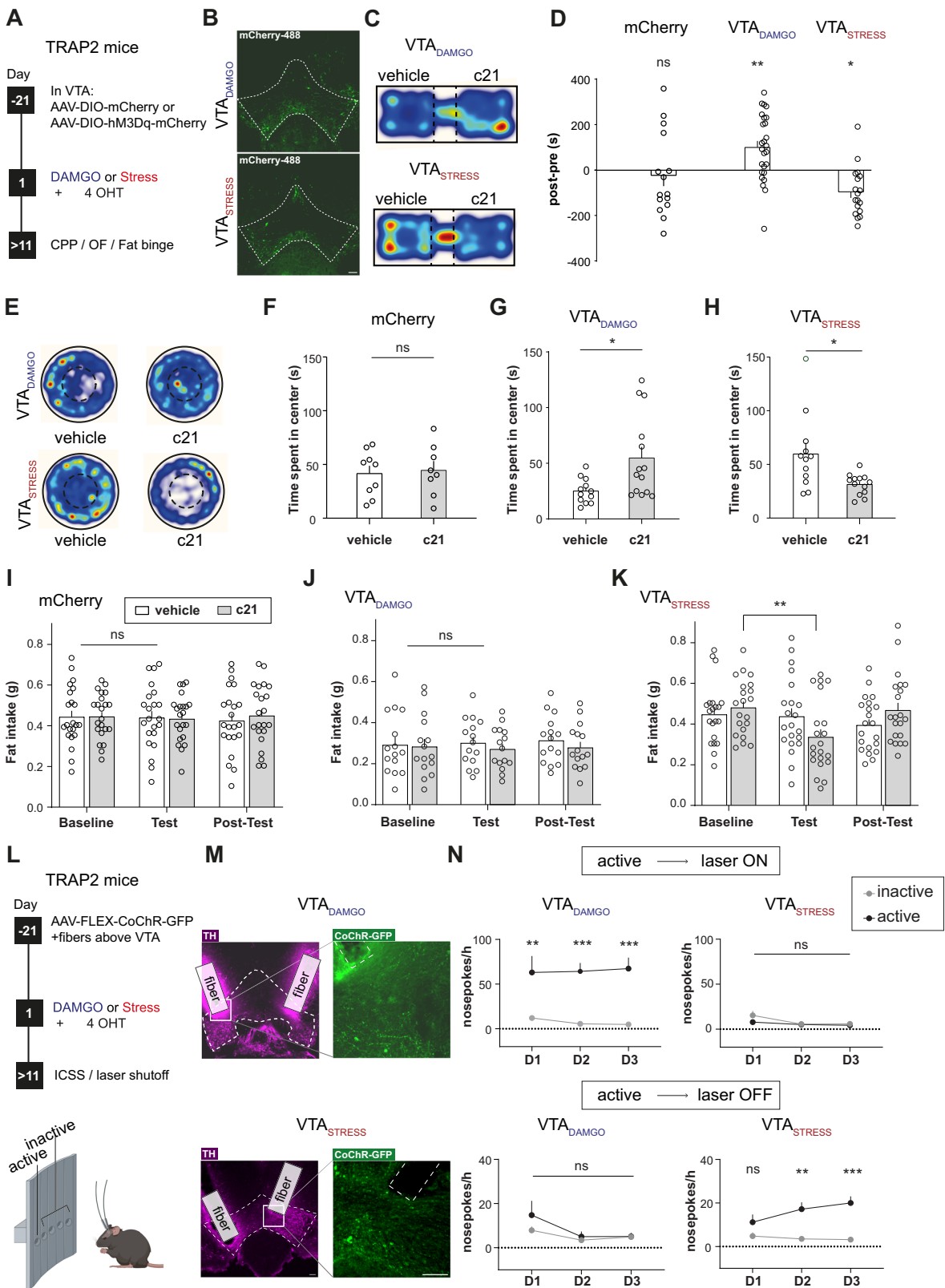

these conditions, nose-poking in the active port resulted in laser deactivation for 20 sec. We found that mice with the opsin in the VTA_STRESS ensemble nose poked significantly more in the active port to pause stimulation, whereas this did not occur for mice with the opsin in the VTA_DAMGO ensemble (Fig. 4N bottom, Suppl. Figure 4J bottom).

Together these data suggest that activation of the VTA_DAMGO ensemble mimics the rewarding and anxiolytic properties of acute

DAMGO administration (see Fig. 1B, F). However, VTA_DAMGO ensemble activation is not sufficient to alter food reward intake, which is heightened by systemic administration of DAMGO (see Fig. 1I). Conversely, stimulation of the VTA_STRESS ensemble is sufficient to recapitulate all the effects of acute social stress exposure (Fig. 1C, G, J). Namely, VTA_STRESS stimulation is aversive, drives anxiety-like behavior, and causes hypophagia. These results establish the potency with which

**Fig. 4 | Stimulation of VTA$_{DAMGO}$ and VTA$_{STRESS}$ ensembles drives congruent approach/avoidance behaviors. A** Experimental timeline. **B** Representative examples of expression of hM3Dq-mCherry in VTA$_{DAMGO}$ or VTA$_{STRESS}$. Scale bar: 100 μm. **C** Representative heatmaps for compartment preference in VTA$_{DAMGO}$ and VTA$_{STRESS}$ groups. **D** Preference score for mCherry controls, VTA$_{DAMGO}$ and VTA$_{STRESS}$ groups ($N_{mCherry}$=15, $N_{DAMGO}$ = 26, $N_{Stress}$ = 18, Two-way RM ANOVA with Time (pre vs post) and Group. Time x Group interaction: $F_{(2,56)}$ = 8.612, $p$ = 0.0005; Sidak's post-hoc for VTA$_{mCherry}$: pre vs post, $t_{(56)}$ = 0.3854, $p$ = 0.9734; VTA$_{DAMGO}$: pre vs post, $t_{(56)}$ = 3.281, $p$ = 0.0053, Sidak's post-hoc for VTA$_{STRESS}$: $t_{(56)}$ = 2.668, $p$ = 0.0296). **E** Representative heatmaps for the open field. **F** Time in center of the open field for mCherry controls ($N_{veh}$=9, $N_{c21}$ = 8, unpaired t test, $t_{(15)}$ = 0.02754, $p$ = 0.7868). **G** As (**F**) for VTA$_{DAMGO}$ ($N_{veh}$=12, $N_{c21}$ = 14, unpaired t test, $t_{(24)}$ = 2.669, $p$ = 0.0134). **H** As (**F**) for VTA$_{STRESS}$ ($N_{veh}$=12, $N_{c21}$ = 13, unpaired t test, $t_{(23)}$ = 2.804, p = 0.01). **I** Fat consumed for mCherry controls during baseline, test, and post-test days (N = 22, Two-way RM ANOVA, Day x Ligand interaction: $F_{(2,84)}$ = 0.2749, $p$ = 0.7603). **J** As (**I**) but for VTA$_{DAMGO}$ group (N = 15, Two-way RM ANOVA, Day x Ligand interaction, $F_{(2,56)}$ = 0.3618, $p$ = 0.698). **K** As (**I**) but for

VTA$_{STRESS}$ group (N = 21, Two-way RM ANOVA, Day x Ligand interaction, $F_{(2,80)}$ = 5.825, $p$ = 0.0043, Tukey's post-hoc for c21: baseline vs test, $q_{(20)}$ = 5.508, $p$ = 0.0025). **L** Experimental timeline for operant behavior. **M** Representative VTA slices showing expression of CoChR-GFP in and fiber traces above the VTA. Scale bar: 100 μm. **N** Top: Nose-pokes to laser stimulate VTA$_{DAMGO}$ or VTA$_{STRESS}$ ensembles ($N_{DAMGO}$ = 9, Two-way RM ANOVA, Nose-poke main effect, $F_{(1,16)}$ = 25.76, $p$ = 0.0001. $N_{STRESS}$ = 10, Two-way RM ANOVA, Day main effect, $F_{(2,32)}$ = 0.06370, $p$ = 0.0128). Bottom: Nose-pokes to deactivate laser stimulation for VTA$_{DAMGO}$ and VTA$_{STRESS}$ ensembles ($N_{DAMGO}$ = 9, Two-way RM ANOVA, Day main effect, $F_{(1.163,18.60)}$ = 6.327, $p$ = 0.0178. $N_{STRESS}$ = 10, Two-way RM ANOVA, Day x Nose-poke interaction, $F_{(2,36)}$ = 4.599, $p$ = 0.0167, Bonferroni post-hoc for D2: inactive vs active, $t_{(56)}$ = 4.248, $p$ = 0.0049, Bonferroni post-hoc for D3, inactive vs active, $t_{(56)}$ = 5.077, $p$ = 0.0008). Mice used for the inactivation of the laser are the same as those used for the activation of the laser previously. Data are presented as mean values + SEM. All statistical tests were performed two-sided. Source data are provided as a Source Data file.

intermingled VTA ensembles can orchestrate opposite approach versus avoidance behavioral patterns.

## Activity of VTA$_{DAMGO}$ and VTA$_{STRESS}$ ensembles is indispensable for reward and stress-driven approach/avoidance behaviors

We showed that stimulating VTA$_{DAMGO}$ and VTA$_{STRESS}$ ensembles recapitulates many of the approach/avoidance behaviors of systemic DAMGO and acute stress itself. However, DAMGO- and stress-induced behavioral effects will involve ensembles across multiple brain regions. Therefore we now asked whether activity in these small ensembles in the VTA were indispensable for the integral DAMGO and stress-driven behaviors to occur. First we established that neither the repeated chemogenetic inhibition of the VTA$_{DAMGO}$ ensemble nor of the VTA$_{STRESS}$ ensemble itself directly affected place preference outcomes (Suppl. Fig. 5A, B). This allowed us to then evaluate the requirement of activity in the ensembles for the integral effects of DAMGO-driven conditioned place preference (CPP) and of stress-induced conditioned place aversion (CPA). For that, mice were injected bilaterally in the VTA with AAV-hSyn-DIO-hM4Di-mCherry or AAV-hSyn-DIO-mCherry as a control. Three weeks post-injection, mice were exposed to either systemic DAMGO or a social stress episode, followed by a 4-OHT injection to enable ensemble tagging, and afterwards behavioral testing was performed (Fig. 5A, B).

For DAMGO-induced place preference experiments, mice expressing mCherry or hM4Di-mCherry in the VTA$_{DAMGO}$ ensemble received c21, followed by DAMGO. Subsequently, mice were randomly placed in one of the CPP compartments. On the following day, mice were placed in the other compartment after c21 injection followed by vehicle. On the third day, mice were allowed to explore the arena freely for 20 minutes (Fig. 5C). We found that inhibiting the VTA$_{DAMGO}$ ensemble during the DAMGO conditioning sessions fully prevented DAMGO-induced CPP (Fig. 5D, E). Similarly, we found that inhibiting the VTA$_{STRESS}$ ensemble during stress conditioning fully prevented stress-driven CPA (Fig. 5D, F). To further establish that this effect is specific to each ensemble, a group of VTA$_{DAMGO}$ mice expressing hM4Di, was subjected to the social stress induced CPA paradigm. We found that inhibiting the VTA$_{DAMGO}$ ensemble during stress conditioning sessions did not block the establishment of stress-induced CPA (Suppl. Figure 5C). Together these data show that activity in these VTA ensembles is selectively necessary for DAMGO or acute social stress-driven conditioned responses.

Finally, we investigated whether VTA$_{DAMGO}$ and VTA$_{STRESS}$ ensemble activity is necessary for the emergence of the effects of DAMGO (increase) or acute social stress (decrease) on palatable food intake. First we established that (in the absence of DAMGO or stress) just chemogenetically inhibiting the VTA$_{DAMGO}$ or the VTA$_{STRESS}$ ensemble did not directly affect fat intake (Suppl. Fig. 5D–G). Then we

tested the effect of ensemble inhibition on the effects exerted by DAMGO or social stress on food intake (Fig. 5G, H). We found that inhibiting the VTA$_{DAMGO}$ ensemble prior to DAMGO administration blocked the resultant hyperphagia, suggesting that activity of the VTA$_{DAMGO}$ ensemble is necessary for DAMGO-induced increase of palatable food intake (Fig. 5I, Suppl. Figure 5H). On the other hand inhibiting the activity of the VTA$_{STRESS}$ ensemble prior to social stress did not prevent the resultant hypophagia (Fig. 5J, Suppl. Fig. 5I–K).

In conclusion, our findings demonstrate that VTA$_{DAMGO}$ and VTA$_{STRESS}$ neuronal ensembles play distinct, essential roles in establishing conditioned place preference and avoidance, respectively. Moreover, the ability of systemic DAMGO to drive fat intake is dependent on activity in the VTA$_{DAMGO}$ ensemble. These findings highlight the importance of these small VTA ensembles in orchestrating the behavioral repertoire in response to the integral experiences driven by external positive and negative valence stimuli.

## Dopaminergic and GABAergic subsets promote approach behavior in the VTA$_{DAMGO}$ ensemble and avoidance behavior in the VTA$_{STRESS}$ ensemble

We next investigated the functional uniformity of different NT-defined neuronal subpopulations within the VTA$_{DAMGO}$ and VTA$_{STRESS}$ ensembles. To this end we used the offspring of TRAP2 mice crossed with either Pitx3-Flp or Vgat-Flp mice. We injected the same mice with intersectional viruses[12,28] for both stimulatory optogenetics (AAV-Conn/Fon-ChR2-EYFP) and inhibitory chemogenetics (AAV-Conn/Fon-hM4Di-mCherry) in the VTA. In this way we targeted either dopaminergic or GABAergic neuronal subsets within the ensemble. An optic fiber was also implanted above the VTA (Fig. 6A, B).

Previously we showed that mice would seek stimulation of their VTA$_{DAMGO}$ ensemble, but not their VTA$_{STRESS}$ ensemble (Fig. 4L–N). Now we sought to address whether stimulation of either the GABAergic or the dopaminergic subset of the VTA$_{DAMGO}$ ensemble would be self-stimulated. For that we exposed mice to either DAMGO i.p. (or to acute social stress) in the presence of 4-OHT, to permanently tag and induce ChR2 expression in the GABAergic (TRAP2xVgat-Flp mice) or in the dopaminergic (TRAP2xPitx3-Flp mice) parts of the VTA$_{DAMGO}$ ensemble (Fig. 6A, B). Using TH immunohistochemistry we determined that TH-positive neurons were indeed preferentially tagged in TRAP2xPitx3-Flp mice (~80% of the tagged ensemble) compared to TRAP2xVgat-Flp mice (~10% of the tagged ensemble) (Fig. 6C, D). We placed the mice in an operant cage to assess their self-stimulation behavior through nose-poking in an active port, as before (Fig. 4L–N). We found that mice sought out the active port significantly more so than the inactive ports, regardless of whether the dopaminergic or GABAergic subset of the VTA$_{DAMGO}$ ensemble was targeted (Fig. 6E, left; Suppl. Figure 6A, left). Mice showed no preference for nose

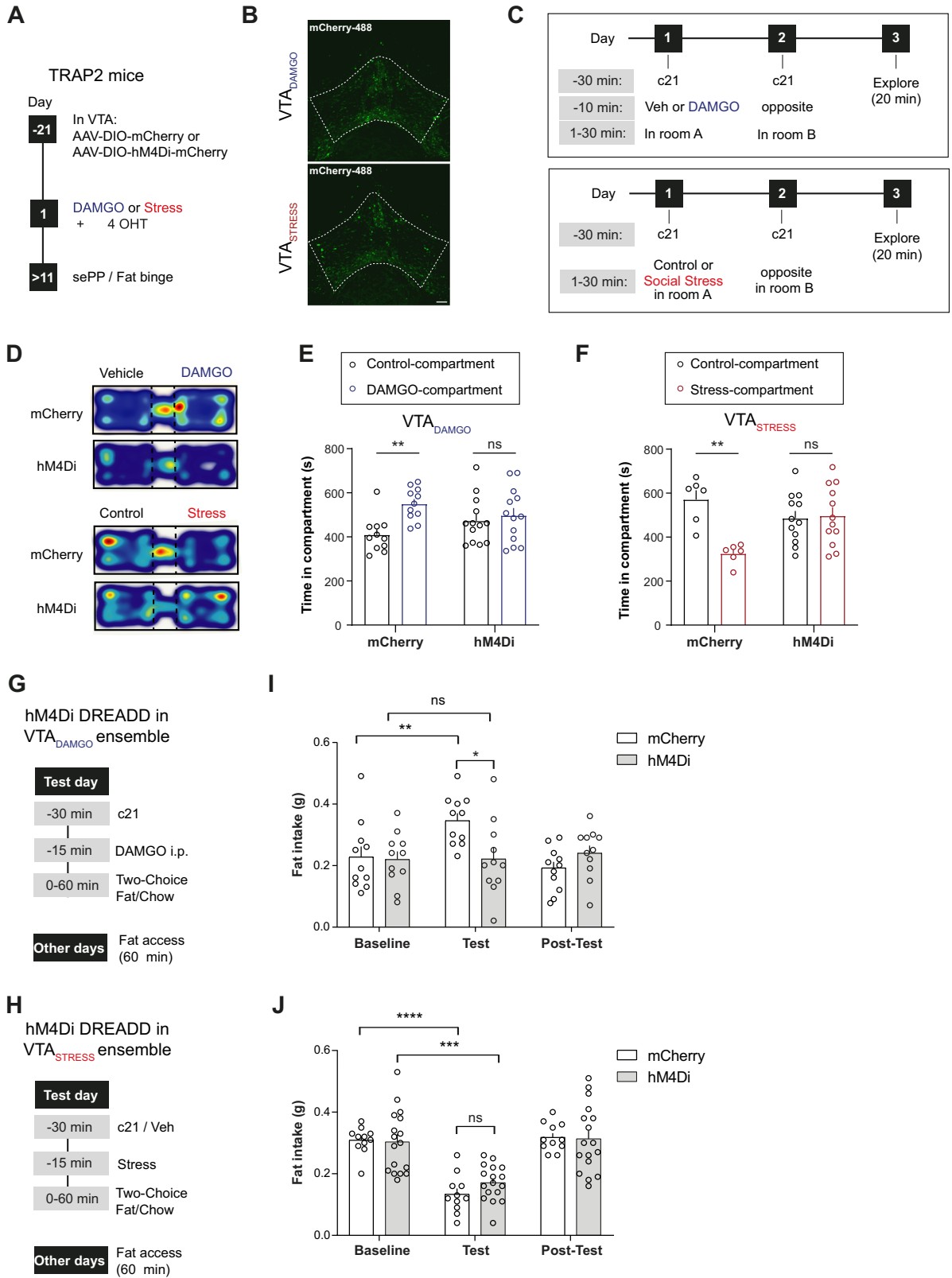

poking to activate the laser for the VTA$_{STRESS}$ ensemble, irrespective of whether the GABAergic or the dopaminergic part was targeted (Fig. 6E, right; Suppl. Figure 6A, right).

Next we switched the contingencies of the experiment. We continuously stimulated the ensemble when the mice were in the operant box. In this case, nose poking in the correct port temporarily turned off laser stimulation. Under these conditions, mice were motivated to turn

off the laser when their stress ensemble (but not their DAMGO ensemble) was stimulated, and this was the case irrespective of whether the dopaminergic or the GABAergic part of the VTA$_{STRESS}$ ensemble was targeted (Fig. 6F; Suppl. Figure 6B).

We next investigated the importance of dopaminergic and GABAergic subsets within the VTA ensembles for the establishment of DAMGO-driven place preference or social stress driven-place

**Fig. 5 | Activity of VTA$_{DAMGO}$ and VTA$_{STRESS}$ ensembles is indispensable for reward and stress-driven approach/avoidance behaviors. A** Experimental timeline. **B** Representative examples of expression of hM4Di-mCherry in VTA$_{DAMGO}$ or VTA$_{STRESS}$. Scale bar: 100 μm. **C** 3-day place preference experimental timeline (**D**) Representative heatmaps showing time spent in vehicle-paired and DAMGO-paired compartments for the VTA$_{DAMGO}$ mCherry or hM4Di groups and control-paired or stress-paired for the VTA$_{STRESS}$ mCherry or hM4Di groups. **E** Time spent in vehicle-paired and DAMGO-paired compartments in VTA$_{DAMGO}$ mCherry- or hM4Di-expressing animals (N$_{mCherry}$=11, N$_{hM4Di}$ = 13, Two-way ANOVA, Compartment x Group interaction F(1,44) = 4.063, $p$ = 0.049, Sidak's post-hoc for mCherry: veh-paired vs DAMGO-paired, t(44) = 3.311, $p$ = 0.0037, Sidak's post-hoc for hM4Di: veh-paired vs DAMGO-paired, t(44) = 0.6217, $p$ = 0.786). **F** Time spent in control-paired or stress-paired compartments in VTA$_{STRESS}$ mCherry- or hM4Di-expressing animals (N$_{mCherry}$=6, N$_{hM4Di}$ = 12, Two-way ANOVA, Compartment x Group interaction, F(1,32) = 10.43, $p$ = 0.0029, Sidak's post-hoc for mCherry: ctrl-paired vs stress-paired, t(32) = 3.782, $p$ = 0.0013, Sidak's post-hoc for hM4Di, ctrl-paired vs stress-

paired, t(32) = 0.2458, $p$ = 0.786). **G** Fat binge experimental timeline after DAMGO administration. **H** Fat binge experimental timeline after acute social stress exposure. **I** Amount of fat consumed for VTA$_{DAMGO}$ mCherry- or hM4Di-expressing animals during baseline, test, and post-test days. During test-day mice received c21 (2 mg/kg) followed by an injection of DAMGO (1 mg/kg) before starting the feeding session (N$_{mCherry}$=11, N$_{hM4Di}$ = 11, Two-way RM ANOVA, Day x Group interaction, F(2,40) = 6.495, $p$ = 0.0036, Sidak's post-hoc for mCherry: baseline vs test, t(10) = 3.867, $p$ = 0.0094, Sidak's post-hoc for hM4Di: baseline vs test, t(10) = 0.04835, $p$ > 0.9999). **J** Amount of fat consumed for VTA$_{STRESS}$ mCherry- or hM4Di-expressing animals during baseline, test, and post-test days. During test-day mice received c21 (2 mg/kg) followed by a 20 sec social stress episode before starting the feeding session (n$_{mCherry}$=11, n$_{hM4Di}$ = 17, Two-way RM ANOVA, Day Main effect F(1.503,39.07) = 54.08, $p$ < 0.0001). Data are presented as mean values + SEM. All statistical tests were performed two-sided. Source data are provided as a Source Data file.

---

avoidance. As stated above, the same mice used for the optogenetic experiment had, aside from an intersectional optogenetic construct, also an intersectional inhibitory DREADD in either the dopaminergic (TRAP2xPitx3-Flp) or the GABAergic (TRAP2xVgat-Flp) parts of their VTA ensembles. Indeed, using TH immunohistochemistry, we established that this strategy preferentially targeted the chemogenetic actuator to TH-positive neurons in TRAP2xPitx3-Flp mice and to TH-negative neurons in TRAP2xVgat-Flp mice (Fig. 6G, H). We subjected the mice to DAMGO-induced sePP or social stress-induced sePA as before (Fig. 5E, F). We found that inhibiting the GABAergic part of the VTA$_{DAMGO}$ ensemble did not prevent the establishment of DAMGO-driven place preference. Instead, when we inhibited the dopaminergic part of the VTA$_{DAMGO}$ ensemble there was no significant preference for the DAMGO-paired compartment (Fig. 6I). In contrast, when examining the effects of inhibiting specific NT-defined subsets in the context of social stress-induced sePA, we found that blocking the activity of either the dopaminergic or the GABAergic part of the VTA$_{STRESS}$ ensemble was sufficient to prevent the establishment of stress-induced conditioned place avoidance (Fig. 6J).

## Discussion

Here we describe that a rewarding and an aversive experience engage two distinct VTA neuronal ensembles, that are intermingled and of largely similar neurotransmitter content, to produce opposite valence behaviors. By combining TRAP2 with Fos immunohistochemistry, we demonstrate that DAMGO and social stress activate distinct neuronal ensembles that are preferentially reactivated by the same experience rather than an experience of contrasting valence. Thus, these ensembles also exhibit a stable stimulus response-selectivity over weeks. Furthermore, using fiber photometry, we show that a VTA$_{DAMGO}$ ensemble is activated in vivo by the intake of palatable food, while a VTA$_{STRESS}$ ensemble is activated by foot shocks, but not food intake. Our chemo- and optogenetic experiments show that the activation of these ensembles can have potent effects on valence, anxiety and in the case of the VTA$_{STRESS}$ ensemble also food intake. In general, these experiments show that activating these small ensembles has strong effects that largely mimic the behavioral repertoires driven by the integral experiences of DAMGO or acute social stress. Importantly, we demonstrate that activity of these ensembles is necessary for the establishment of conditioned place preference/avoidance. Activity in the VTA$_{DAMGO}$ ensemble is also critical for opioid-driven fat consumption. Finally, we show that within the VTA$_{DAMGO}$ and VTA$_{STRESS}$ ensembles, different neurotransmitter-defined neuronal subsets (e.g., GABAergic and dopaminergic neurons), serve valence-related functions that are in accordance with the role of the given ensemble as a whole, rather than presenting opposite valence functions within the ensemble. These findings highlight that even small ensembles in the VTA (i.e., about 10% of VTA neurons each) are critical not only for

reward-triggered reinforcing behaviors, but also for the manifestation of stress-driven behavioral avoidance.

## VTA ensembles responding to valence stimuli

Our study highlights that VTA$_{DAMGO}$ and VTA$_{STRESS}$ ensembles are to a considerable extent non-overlapping subsets, preferentially activated by congruent valence stimuli over non-congruent valence stimuli. VTA$_{DAMGO}$ cells were, even weeks later, again preferentially activated by DAMGO over acute social stress. Moreover, these cells were also active during food reward approaches. VTA$_{STRESS}$ cells instead were, again even weeks later, preferentially activated by social stress over DAMGO. Moreover, they did not exhibit calcium reactivity during palatable food intake, whereas they were strongly activated during electric shock. These findings suggest that these two ensembles have selectivity in responding to either a congruently positive or negative experience. Instead, their responses are not limited to a singular type of experience (e.g., the VTA$_{STRESS}$ ensemble responds to different types of stressors, and the VTA$_{DAMGO}$ ensemble to different types of rewards). This is an interesting observation in the context of pure versus mixed selectivity response patterns of neurons[29]. We show here that the VTA functional organization recruits specific ensembles that, weeks later, keep responding with high selectivity to multiple preferentially congruent valence stimuli.

We found that the VTA$_{STRESS}$ and VTA$_{DAMGO}$ ensembles were not clearly topographically segregated. It is of note that for the VTA at large, there is evidence for at least partial topographical organization in terms of circuit connectivity, and in terms of the general role of VTA sub-territories in behavioral functions[9,30]. For instance with, on average, dopamine neurons in lateral parts of the VTA being more linked to reward-related processes, and medial parts of the VTA being more linked to aversion-related processes, though these are not absolute divisions in functionality[9,30]. Our results, based on Fos-based methods rather than for instance in vivo electrophysiological responses, indicate that VTA neurons with opposite roles in valences processes are at least in part intermingled in similar VTA sub regions. These findings do leave open the possibility of topographical gradients in the VTA, in terms of where it may be more likely to find neurons with certain valence properties, especially when assessed with other methods (e.g., in vivo electrophysiology).

We showed that two congruent valence stimuli resulted in more overlap in a given VTA ensemble than did incongruent valence stimuli. However, even with two incongruent valence stimuli, ~15% of VTA neurons showed TRAP/Fos overlap. There are multiple possible contributing factors to this amount of overlap with incongruent valence stimuli. It could reflect a general baseline of Fos-based activity. We did indeed find that this value of overlap was not larger than what we observed with two subsequent home cage experiences. Nevertheless, a 4OH-Tamoxifen i.p. injection in a home cage likely does also represent

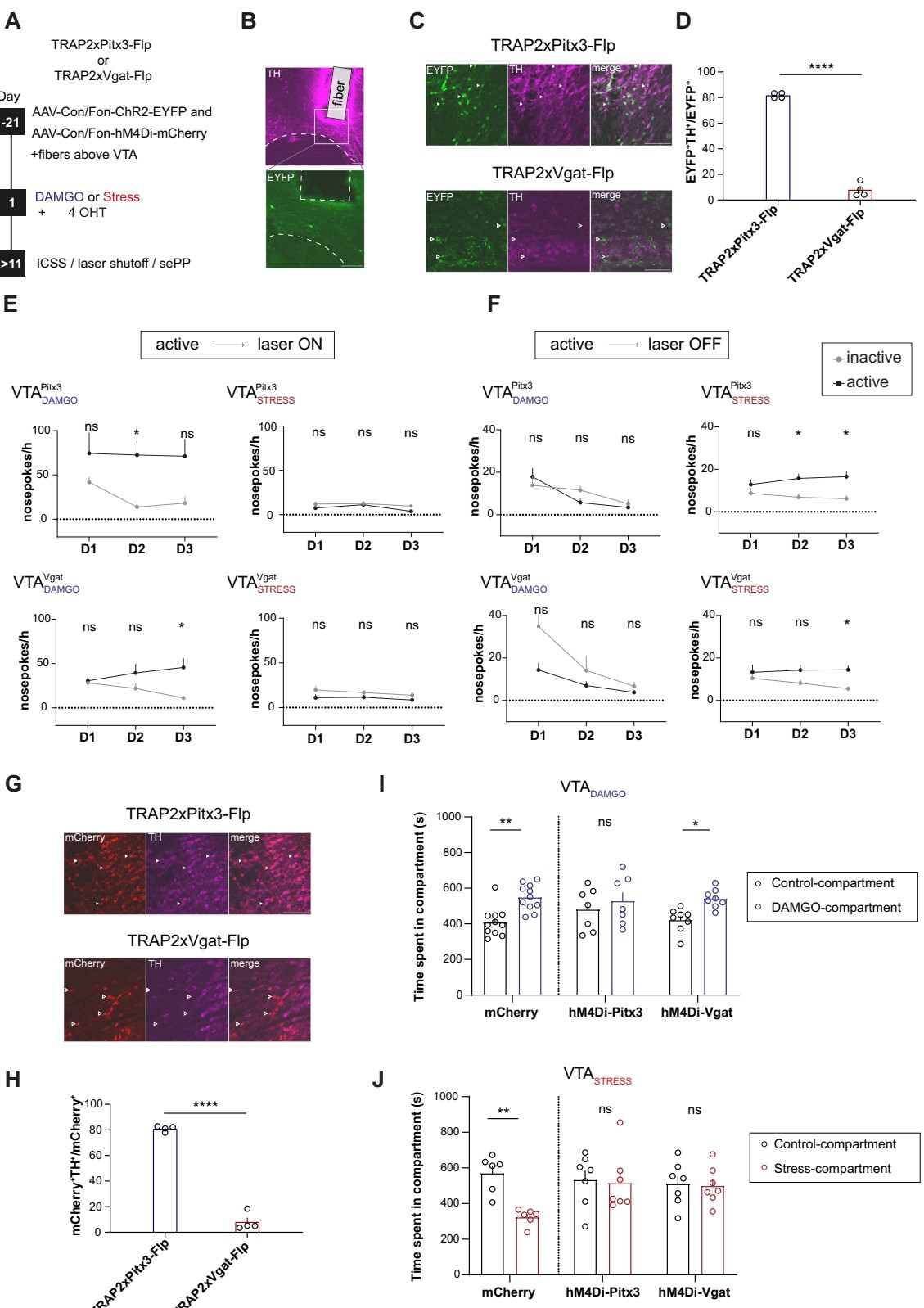

an experience, rather than being fully neutral. It is therefore also possible that the extent of overlap in Fos in VTA ensemble neurons with incongruent valence stimuli represents features of salience of the stimuli. Salience/arousal responses (e.g., responses to both negative and positive valence) have indeed been proposed to be represented by VTA neurons[14,31,32]. For instance, in a recent study, Willmore and colleagues using calcium imaging showed that ~2% of VTA_DOPAMINE

neurons are activated by both food and an aversive air-puff[17]. Furthermore, del Arco and colleagues using in vivo electrophysiology, showed that in the VTA ~ 25% of neurons are activated by both a food reward and a tail pinch[33]. Salience/arousal encoding neurons may therefore be partially present in both the VTA ensembles we describe here. Another consideration is that after a stressor subsides, the subsequent relief period may also be in itself a rewarding experience,

**Fig. 6 | Dopaminergic and GABAergic parts of VTA$_{DAMGO}$ and VTA$_{STRESS}$ ensembles are functionally uniform. A** Experimental timeline. **B** Representative fiber placement over VTA. Scale bar: 100 μm. **C** Representative images of ChR2-EYFP and TH in VTA for TRAP2- Pitx3-Flp (top) and TRAP2-Vgat-Flp (bottom). White arrowheads: co-localized EYFP/TH. Open arrowheads: EYFP-positive, TH-negative neurons. Scale bar: 100 μm. **D** Proportion TH$^+$ ensemble neurons across groups (N = 4/group, unpaired t test, t(6) = 25.57, $p < 0.0001$). **E** Top: Nose pokes to laser stimulate the dopaminergic part of the ensembles (N$_{DAMGO}$ = 7, Two-way RM ANOVA, Nose poke main effect: F(1,12) = 6.231, $p = 0.0281$. N$_{STRESS}$ = 7, Two-way RM ANOVA, Day main effect, F(1.646, 19.76) = 4.129, $p = 0.0383$). Bottom: Nose pokes to stimulate the GABAergic part of the ensembles (N$_{DAMGO}$ = 9, Two-way RM ANOVA, Nose poke x Day interaction, F(2,32) = 5.006, $p = 0.0128$, Bonferroni post-hoc for D3, inactive vs active, t(8.459) = 3.358, $p = 0.0276$. N$_{STRESS}$ = 7, Two-way RM ANOVA, Nose poke main effect, F(1,12) = 4.669, $p = 0.0516$). **F** Top: Nose-pokes to deactivate stimulation of the dopaminergic part of the ensembles (N$_{DAMGO}$ = 7, Two-way RM ANOVA, Day main effect, F(1.617, 19.41) = 12.86, $p = 0.0005$. N$_{STRESS}$ = 7, Two-way RM ANOVA, Nose poke main effect, F(1,12) = 14.63, $p = 0.0024$. Bottom: Nose-pokes to deactivate stimulation of the GABAergic part of the ensembles (N$_{DAMGO}$ = 9, Two-

way RM ANOVA, Nose-poke main effect, F(1,16) = 4.526, $p = 0.0493$. N$_{STRESS}$ = 7, Two-way RM ANOVA, Nose-poke main effect, F(1,12) = 6.958, $p = 0.0217$.
**G** Representative images of hM4Di-mCherry and TH in VTA for TRAP2-Pitx3-Flp (top) and TRAP2-Vgat-Flp (bottom). White arrowheads: colocalized mCherry/TH. Open arrowheads: mCherry-positive TH-negative neurons. Scale bar: 100 μm. **H** Proportion TH$^+$ ensemble neurons across groups (N = 4/group, unpaired t test, t(6) = 19.98, $p < 0.0001$). **I** Time in vehicle-paired and DAMGO-paired compartments in VTA$_{DAMGO}$ mCherry-expressing animals (data as in Fig. 5E), hM4Di-Pitx3-TRAP2, or hM4Di-Vgat-TRAP2 animals (N$_{mCherry}$=11, N$_{Pitx3}$ = 7, N$_{Vgat}$ = 8, Two-way ANOVA, Compartment main effect, F(1,46) = 16.60, $p = 0.0002$. **J** Time in control-paired or stress-paired compartments in VTA$_{STRESS}$ mCherry-expressing animals (data as in Fig. 5F), hM4Di-Pitx3-TRAP2 or hM4Di-Vgat-TRAP2 animals (N$_{mCherry}$=6, N$_{pitx3}$ = 7, N$_{Vgat}$ = 7, Two-way ANOVA, Compartment x Group interaction, F(2,34) = 3.74, $p = 0.034$, Sidak's post-hoc for ctrl-paired vs stress-paired in mCherry mice: t(34) = 3.469, $p = 0.0043$; hM4Di-Pitx3-TRAP2 mice: t(34) = 0.2631, $p = 0.9913$, and hM4Di-Vgat-TRAP2 mice: t(34) = 0.1704, $p = 0.9976$. Data are presented as mean values + SEM. Statistical tests performed two-sided. Source data are provided as a Source Data file.

transiently engaging dopamine neurons in the VTA[34]. Our VTA$_{STRESS}$ ensemble could therefore have also contained 'stress-relief' neurons. However, our ensemble stimulation and inhibition experiments showed clear net reinforcing properties of the VTA$_{DAMGO}$ ensemble and clear net avoidance properties of the VTA$_{STRESS}$ ensemble. Overall this suggests that we did mainly capture valence rather than salience aspects when tagging the ensembles, and mainly positive valence in the VTA$_{DAMGO}$ and negative valence in the VTA$_{STRESS}$ ensemble.

## The importance of small VTA neuronal ensembles and their necessity for behavioral effects

A remarkable outcome of this study is that activity in small VTA ensembles is crucial for the approach and avoidance behaviors driven by systemic opioid and acute social stress stimuli. These are experiences that engage many brain regions, also other than VTA[35–38]. Moreover, the VTA is a small brain region in itself[39]. We report that based on Fos immunostaining the VTA ensembles are ~10% of VTA neurons (based on NeuN$^+$ cells in VTA). Notably, using the TRAP2 technique we do not fully capture this population. As we showed before, TRAP2 counts of VTA cells are approximately a third of the population labeled by endogenous Fos[23]. Extrapolating from this, we therefore estimate, also keeping incomplete targeting via viral approaches in mind, that the manipulated VTA ensembles in this study likely amount to ~2–6% of all VTA neurons. Stereological estimates of neuron count in the mouse VTA, suggest that it harbors ~20–30k dopamine neurons, and also about 20k non-dopaminergic neurons, and thus about 50k neurons in total[39]. Consequently, our data suggest that we are manipulating ~1000–3000 neurons within either the VTA$_{DAMGO}$ or the VTA$_{STRESS}$ ensemble. Stimulation of this small amount of neurons is thus able to drive effects on place preference, anxiety behavior and, in the case of the VTA$_{STRESS}$ ensemble, also food intake. Yet more strikingly even, the inhibition of just this small amount of neurons shows their indispensability in mediating certain behavioral effects. Specifically, for systemic DAMGO, the resultant place preference and hyperphagia, and for acute social stress the place aversion, but not its hypophagic effects.

The potency of these two small ensembles in shaping behavioral responses may in part stem from the fact that they are each comprised of ~50% of dopamine neurons. Dopamine neurons, through their complex axonal arborizations with many release sites, allow neuromodulatory effects over large areas of target substrates[40]. Furthermore, neuronal ensembles in other regions than VTA have also been reported to typically comprise small clusters of cells, often less than 10% of neurons. For instance, inhibition of about 6% of neurons in the medial prefrontal cortex can suppress drug relapse[41], and stimulation of as few as 800 AgRP neurons in the arcuate hypothalamus can drive

food intake[42]. In another study, direct optogenetic stimulation of small ensembles of basolateral amygdala neurons, with either sucrose or quinine responsivity, could alter consummatory behavior[43]. Together these findings emphasize the increasing recognition of the power of small neuronal ensembles in shaping behavior.

Our experiments also addressed the extent to which different neurotransmitter-defined neuronal cell types within a VTA ensemble: (i) have similar or rather opposite roles in valence processes and associated behavioral responses, (ii) and whether it is the ensemble as a whole, or the activity of some of its key constituents that is important for approach/avoidance behaviors. We show that mice seek to turn off optogenetic stimulation of either the GABAergic or the dopaminergic subset of its VTA$_{STRESS}$ ensemble, while they will instead seek to turn on such stimulation of either the GABAergic or the dopaminergic subset of their VTA$_{DAMGO}$ ensemble. This finding highlights that there are sets of dopaminergic and GABAergic neurons with opposite valence functions, which within a given ensemble may have similar functions with regards to orchestrating approach or avoidance behaviors. Moreover, with our chemogenetic inhibition experiments we evaluated whether activity of both the GABAergic and the dopaminergic subsets within the VTA$_{DAMGO}$ and VTA$_{STRESS}$ ensembles were equally crucial for the emergence of DAMGO-driven approach, and social stress-driven avoidance behavior, respectively. Here we observed differences in this regard between the ensembles. Chemogenetic inhibition of the VTA$_{DAMGO}$ ensemble as a whole blocked DAMGO-driven approach behavior. Chemogenetic inhibition of just the dopaminergic subset of this ensemble still attenuated approach behaviors, but inhibition of just the GABAergic subset of the VTA$_{DAMGO}$ ensemble did not. This indicates that for the VTA$_{DAMGO}$ ensemble, its NT-defined sub-constituents may have seemingly similar valence functions when stimulated, but the activity of each of its NT-defined subsets is not equally indispensable for the emergence of DAMGO-driven approach behavior as a whole. A partially different image emerged for the VTA$_{STRESS}$ ensemble. Chemogenetic inhibition of the VTA$_{STRESS}$ ensemble as a whole blocked social stress-driven avoidance. This also occurred when we chemogenetically inhibited only the GABAergic or only the dopaminergic subsets of the VTA$_{STRESS}$ ensemble. This indicates that the GABAergic and dopaminergic neuronal types within the VTA$_{STRESS}$ ensemble have similar valence functions when stimulated, and that the activity of both of these subsets is critical for social stress to drive avoidance behavior.

These findings also raise interesting questions on whether different neuronal cell types within the VTA ensembles are locally connected to each other, and whether ensembles as a whole are interconnected. Based on our findings we predict that, whereas it is known that VTA$_{GABA}$ neurons have a certain degree of local

connectivity with VTA$_{DOPAMINE}$ neurons[6,44], it may be more likely that VTA$_{GABA}$ neurons from within a VTA$_{STRESS}$ or VTA$_{DAMGO}$ ensemble do not preferentially innervate VTA$_{DOPAMINE}$ neurons within the same ensemble. Future research will need to address this.

### Divergence in terms of mimicry and prevention for the two VTA ensembles for fat intake

We show a dissociation for the role of VTA$_{DAMGO}$ and VTA$_{STRESS}$ ensembles regarding fat intake. We recapitulate that systemic DAMGO causes hyperphagia, in accordance with previous reports[45–47], but we also find that solely stimulating only the VTA$_{DAMGO}$ ensemble is not sufficient to mimic this. One possibility in this regard is that VTA$_{DAMGO}$ ensemble activity may contribute to hyperphagia, but only when in conjunction with other network activity (which we then did not recreate by solely stimulating these neurons). Indeed, there are multiple brain regions in which DAMGO can increase fat intake, including not only the VTA, but also the prefrontal cortex, amygdala and nucleus accumbens[46,48,49]. In further support of a contribution of the VTA$_{DAMGO}$ ensemble to palatable food intake we show, with fiber photometry, that VTA$_{DAMGO}$ cells are reactive during fat consumption. Moreover, we show that inhibiting the VTA$_{DAMGO}$ ensemble, during systemic DAMGO, prevents the opioid from increasing fat intake. Our results therefore suggest that while VTA$_{DAMGO}$ activity alone is not enough for increased fat intake, but activity in this ensemble represents a required hub in a larger co-activated network for opioid-driven hyperphagia.

In contrast to the finding for the VTA$_{DAMGO}$ ensemble in hyperphagia, we observed an opposite behavioral pattern for the involvement of the VTA$_{STRESS}$ ensemble in hypophagia. In the acute period after systemic social stress there is a hypophagic response, as we report here and previously[50]. This is in contrast to the hyperphagia that can occur hours to days after the stress[50,51]. We show here that the acute stimulation of the VTA$_{STRESS}$ ensemble is sufficient to (partly) reduce fat intake. However, we also observed that inhibiting these cells alone was insufficient to prevent social stress-driven hypophagia. This suggests that during this acute period of social stress, there are other parallel mechanisms at play rather than just VTA$_{STRESS}$ ensemble activity alone, which together are sufficiently able (also without the contribution of the VTA$_{STRESS}$ ensemble) to deprioritize food reward intake over other potentially adaptive behaviors during such a threatening stressor.

Together our results show that the VTA$_{DAMGO}$ and VTA$_{STRESS}$ ensembles are largely distinct collectives of neurons, that despite showing similarities at the neurotransmitter (and anatomical) level, have opposite net roles in approach and avoidance behavior. Furthermore, despite their small number, they are indispensable hubs for specific approach and avoidance behaviors in response to potent external stimuli with integral effects. The power of targeting these small ensembles may inform the development of more effective interventions for diseases in which valence processes are perturbed.

## Methods

### Animals

Adult male mice (25–35 g, >6 weeks) were used in all experiments. C57BL/6 J (Jax #664), heterozygous vesicular GABA transporter-Cre (vgat-Cre; Jax #016962), heterozygous vesicular glutamate transporter 2-Cre (vglut2-Cre; Jax #28863), heterozygous TRAP2 (Jax #030323), heterozygous Ai14 tdTomato reporter (Jax #007914), heterozygous Pitx3-Flp (generated by the ETH Phenomics Center (EPIC) of ETH Zürich, Switzerland), heterozygous vgat-Flp (Jax #029591) and double heterozygous TRAP2xAi14, TRAP2xPitx3-Flp, and TRAP2xVgat-Flp offspring were bred in house. For the social stress paradigm, proven breeder Swiss-CD1 mice were purchased from Janvier (France) and were used as aggressors. C57BL/6 J, vgat-Cre and vglut2-Cre mice were group-housed (2–4 per cage) in a temperature- and humidity-controlled room (22 ± 2 °C and 60–65% respectively) under a 12 h light/dark cycle (lights on at 7am) with ad libitum access to water and standard laboratory chow [Special Diet Services [SDS], product code CRM(E)]. TRAP2, TRAP2xAi14, TRAP2xPitx3-Flp and TRAP2xVgat-Flp mice were group-housed (2-4 per cage) until 1 week prior to TRAPing after which they were individually housed under the same conditions as described above. All experiments were approved by the Animal Ethics Committee of Utrecht University, and were conducted in agreement with Dutch law (Wet op de Dierproeven, 2014) and European regulations (Guideline 86/609/EEC).

### Stereotactic surgery

Mice were anesthetized with ketamine (75 mg/kg i.p.; Narketan, Vetoquinol) and dexmedetomidine (1 mg/kg i.p.; dexdomitor, Vetoquinol). For local anesthesia, Lidocaine (0.1 ml; 10% in saline; B. Braun) was topically applied under the skin on the skull before incision. Eye ointment cream (CAF, Ceva Sante Animale B.V., Naaldwijk, Netherlands) was applied before surgery. Mice were mounted on a stereotactic frame (UNO B.V., Zevenaar, Netherlands, Model 68U801 or 68U025) and kept on a heating pad (33 °C) during surgery. Injections were done using a 31 G metal needle (Coopers Needleworks, Birmingham, United Kingdom) attached to a 10 µl Hamilton syringe (model 801RN) via flexible tubing (PE10, 0.28 mm ID, 0.61 mm OD, Portex, Keene, NH, United States). The Hamilton syringe was controlled by an automated pump (UNO B.V., Zevenaar, Netherlands, -model 220). For fiber implantation, the skull surface was gently scratched with a scalpel, followed by application of 35% phosphoric acid (167-CE, ultra-Etch, ultradent, USA) for 5 minutes to roughen the surface at the start of surgery. For viral infusions, all mice were bilaterally injected in the VTA (−3.2 mm posterior to bregma, 1.60 mm lateral, −4.90 mm ventral under an angle of 15°). After infusion of the viral volume, needles were left in place for an additional 9.5 min. Then they were moved 50 µm dorsally and completely retracted 30 s later. The skin was subsequently sutured (V926H, 6/0, VICRYL, Ethicon). Mice were then subcutaneously injected with atipamezole (50 mg/kg; Atipam, Dechra), carprofen (5 mg/kg, Carporal) and 1 ml of saline and were left to recover on a heating pad at 36 °C. Carprofen (0.025 mg/l) was provided in drinking water for a week after surgery. Animals were solitarily housed for 3 days post-surgery. Mice were allowed at least 3 weeks prior to subsequent behavioral manipulations.

**For visualizing vgat+ and vglut2+ VTA populations.** vgat-Cre and vglut2-Cre mice were bilaterally injected in the VTA with 0.3 µl of rAAV-hSyn-DIO-mCherry ($4.2 \times 10^{12}$ gc/ml; Addgene) per hemisphere at a rate of 0.1 µl/min to visualize VTA GABAergic and glutamatergic subpopulations, respectively. Specifically for VTA$_{GABA}$ and VTA$_{GLU}$ cell type identification in the VTA$_{STRESS}$ ensemble we used the data from our previous study[23].

**For Fiber Photometry experiments.** TRAP2 mice were bilaterally injected in the VTA with 0.3 µl of AAV-syn-FLEX-jGCaMP8s ($4.2 \times 10^{12}$ gc/ml, Addgene). Optic fibers (400 µm, 0.50NA; FP400URT, Thorlabs) were bilaterally implanted above the VTA (−3.2 mm posterior to bregma, 1.60 mm lateral, −4.75 mm ventral under an angle of 15°) inserted into a ceramic ferrule (bore 430-440 µm; MM-FER2006-4300, Precision Fiber Products). Bilateral placement was performed to increase probability of having one side with measurable calcium dynamics. In order to reinforce stability, four M1,2 × 2 mm screws (Fabory) were secured into the skull at two full rotations each, and a layer of adhesive luting cement (C&B Metabond; Parkell, Edgewood, NY, USA) was added. The cap securing the optic fiber was made with glass ionomer cement (GC Fuji PLUS capsules yellow, Henry Schein). Mice were given a minimum of three weeks to recover before any further procedures were conducted.

**For chemogenetic stimulation/inhibition experiments.** TRAP2 mice were bilaterally injected in the VTA with 0.3 µl of rAAV-hSyn-DIO-hM3Dq-mCherry ($5 \times 10^{12}$ gc/ml; Addgene), rAAV-hSyn-DIO-hM4Di-mCherry ($4.2 \times 10^{12}$ gc/ml; Addgene) or rAAV-hSyn-DIO-mCherry ($4.2 \times 10^{12}$ gc/ml; Addgene) per hemisphere at a rate of 0.1 µl/min. 4 animals from the VTA$_{DAMGO}$ hM3Dq group and 3 animals from the VTA$_{STRESS}$ hM3Dq group were excluded due to misplaced injections.

To specifically express an inhibitory DREADD in the dopaminergic or GABAergic part of the VTA$_{DAMGO}$ or VTA$_{STRESS}$ ensembles, TRAP2xPitx3-Flp or TRAP2xVgat-Flp mice were bilaterally injected in the VTA with 0.3 µl of AAV-nEF-Con/Fon-hM4Di-mCherry ($5 \times 10^{12}$ gc/ml; Addgene) per hemisphere at a rate of 0.1 µl/min.

**For in vivo optogenetic experiments.** TRAP2 mice were bilaterally injected with 0.3 µl of AAV-Syn-FLEX-CoChR-GFP ($2.9 \times 10^{12}$ gc/ml, UNC Vector Core) into the VTA. Optic fibers (200 µm, 0.22 NA, FG105UCA, ThorLabs, Newton, NJ, USA) inserted in a ceramic stick ferrule (0.127 – 0.131 µm bore, CFLC128-10, ThorLabs, Newton, NJ, USA) were implanted above the VTA (− 3.2 mm posterior to bregma, 1.60 mm lateral, −4.70 mm ventral under an angle of 15°). In order to enhance the structural stability, two M1,2 × 2 mm screws were secured into the skull with two full rotations each, and a layer of self-adhesive resin cement (RelyX Unicem Aplicap, Henry Schein) was applied. The cap securing the optic fiber was fabricated using glass ionomer cement (GC Fuji PLUS capsules yellow, Henry Schein). Mice were allowed a minimum recovery period of three weeks before any further procedures were conducted.

To specifically express an excitatory opsin in the dopaminergic or GABAergic part of the VTA$_{DAMGO}$ or VTA$_{STRESS}$ ensembles, TRAP2x-Pitx3-Flp or TRAP2xVgat-Flp mice were bilaterally injected in the VTA with 0.3 µl of AAV-nEF-Con/Fon-ChR2-EYFP ($5 \times 10^{12}$ gc/ml; Addgene) per hemisphere at a rate of 0.1 µl/min and optic fibers were implanted as described above.

**Behavioral paradigms**

**DAMGO i.p. administration.** Mice were habituated to injections by receiving saline i.p. injections daily starting 4 days before DAMGO injection. Mice were left undisturbed the day before DAMGO injection. Then mice were injected with DAMGO i.p. at a dose of 1 mg/kg dissolved in saline in their home cage. Control animals received a saline injection (i.p.) of an equivalent volume. For Fos immunohistochemistry experiments C57BL/6 J mice were perfused 2 h after DAMGO injection. For TRAP2 and TRAP2xAi14 mice, mice were solitarily housed 7 days prior to DAMGO injection. 2 h after the DAMGO/saline injection, mice were injected with 4-hydroxytamoxifen (4-OHT). 4-OHT (Sigma-Aldrich Chemie N.V, Zwijndrecht, Netherlands, H6278) was dissolved in an aqueous solution as described before[23,52]. The final solution contained 2.5 mg/ml 4-OHT, dissolved in 5% DMSO, 1% Tween-80 and saline and was always injected at a dose of 25 mg/kg. For TRAP2xAi14 ensemble overlap experiments, mice were left undisturbed for 3 weeks and were later exposed to experiences of congruent or incongruent valence followed by transcardial perfusion (90 min after onset of social stress, or 120 min after onset of DAMGO or food reward). Congruent valence experiences included a second DAMGO i.p. injection or free access (2 h) to a piece of beef fat (Blanc de Boeuf, Vandermoortele, 9.0 kcal/g) after overnight (13 h) food restriction. Social stress was used as an experience of incongruent valence.

**Social stress paradigm.** Swiss-CD1 male mice were housed individually in a Makrolon cage (type IV, Tecniplast, Buguggiate, Italy). An intruder experimental mouse was introduced in the cage between 8:00–10:30 am. Fighting was tracked live and animals were allowed to fight for an accumulated 20 s. Next, they were separated by a perforated transparent splitter that prevented physical, but allowed sensory interaction for the remaining time of the experiment. For Fos

immunohistochemical experiments, animals remained in the CD1 aggressor cage until perfusion (90 min later). Control animals were placed in a Makrolon cage with a novel male C57BL/6 J conspecific separated by a transparent splitter and were perfused 90 min later.

For TRAP2xAi14 and TRAP2 experiments, animals were solitarily housed 7 days before the stress episode. During this period, mice were habituated to intraperitoneal (i.p.) injections by receiving a saline injection i.p., 4 and 5 days after the start of their solitary housing. During the social stress paradigm, TRAP2xAi14 mice were put in the resident cage and were allowed to fight for an accumulated total of 20 s. Afterwards they remained in the resident cage, separated by a perforated transparent splitter until 4-OHT injection was administered. 4-OHT was administered 3 h after the onset of the fight period. For TRAP2xAi14 ensemble overlap experiments, all mice were placed back in their home cage (solitarily housed) and 3 weeks later were exposed to an experience of congruent (social stress or a 30 min session of foot shocks, during which a total of 20 shocks of 1 s duration, and 0.7 mA intensity was administered with 90 second inter-shock intervals), or incongruent (DAMGO i.p.) valence stimuli. This was followed by transcardial perfusion (for foot shock or social stress 90 min after onset of the stimulus, for DAMGO 120 min after onset of the stimulus).

Three independent groups of TRAP2xAi14 mice received 4-OHT in the absence of a rewarding or aversive experience (home cage controls). These mice were divided into 3 groups and 3 weeks later they were perfused after exposure to DAMGO, social stress, or no additional experience (home cage).

**Conditioned place preference/avoidance.** Conditioned place preference/avoidance (CPP/CPA) experiments were performed in a custom-made 3-compartment rectangular apparatus (74 × 25 cm, 30 cm high) with an open ceiling, for an unforced-choice behavioral paradigm. The middle, neutral zone-compartment separated the other two conditioning zone-compartments, which were distinguishable in wall pattern and floor texture. One test compartment had vertical white-and-black-striped walls and metal grid flooring in square pattern (squares were 0.2 × 0.2 cm) whereas the other test compartment had black walls and metal grid flooring in square patterns (squares were 0.5 × 0.5 cm). The middle compartment had a white wall color. On the conditioning day, two off-white Plexiglas barriers (24 × 40 cm) were placed bordering the neutral zone so that each compartment was closed off. On pre-test and test days, the barriers were removed so that the mice could roam freely between the compartments. Movement was tracked with Ethovision software (Version 11, Noldus Information Technology, the Netherlands).

**For DAMGO single exposure place preference (sePP) experiments.** Following an unbiased approach, on conditioning day 1 of the experiment mice were either placed in the black or striped compartment 10 min after a vehicle or DAMGO (1 mg/kg, i.p.) injection where they were allowed to roam freely for 30 minutes. On conditioning day 2, the mice were placed in the other compartment 10 min after a vehicle or DAMGO (1 mg/kg, i.p.) injection in a counter-balanced manner and were allowed to roam freely for 30 minutes. On day 3, CPP was tested by placing the mice in the neutral zone with the barriers removed and tracking the time spent in the different compartments for 20 min.

**For acute social stress single exposure place avoidance (sePA) experiments.** On conditioning day 1 of the experiment mice were either placed in the black or striped compartment and were allowed to roam freely for 30 minutes. On conditioning day 2, mice were placed in the other compartment, initially undisturbed for 5 min, after which an aggressor was introduced in the compartment. After an accumulated 20 s of fighting was completed, or when 20 min elapsed, the aggressor

was removed from the compartment. The mice remained in the compartment up until a total of 30 min elapsed before placed back to their home cage. Mice were exposed to the CD1 aggressor on day 1 or 2 in a counterbalanced manner. On day 3, CPA was tested by placing the mice in the neutral zone with the barriers removed and tracking the time spent in the different compartments for 20 min.

**For chemogenetic activation/inhibition-paired place preference.** On conditioning day 1, TRAP2 mice were placed in the central compartment and were let to explore freely for 20 min (pre-test). Time spent in each compartment was recorded. On days 2–4, mice received an i.p. injection of saline or c21 between 7–9 am (hM3Dq: 1 mg/kg, hM4Di: 2 mg/kg) 15 min prior to being placed in the CPP compartment and were left there for a total of 30 min. Between 4–6 pm mice received again saline or c21 in a counterbalanced manner and were placed in the other compartment for a total of 30 min. On day 5, mice were placed again in the central compartment and were allowed to explore freely for 20 min. Preference score was calculated as time spent in c21-paired compartment – time spent in vehicle-paired compartment.

**For sePP/sePA necessity experiments in VTA ensemble mCherry- and hM4Di-expressing mice.** On both conditioning days, all animals received an i.p. c21 injection with a dose of 2 mg/kg 15 min prior to DAMGO injection or acute stress exposure. The rest of the sePP/sePA procedure was conducted as described above.

**Open field.** Open-field experiments were executed in a custom-made open field arena which consisted of a square field (87 by 87 cm) and circular wall (251 cm in circumference, 35 cm high). The field of the arena contained white-lined patterns to delineate the center zone (56 cm in circumference) and the peripheral zone (outer border: 251 cm in circumference, inner border; 207 cm in circumference). Ethovision software (Version 11, Noldus Information Technology, the Netherlands) was used to extract movement parameters (distance moved, time in zones, immobility). Immobility was defined as a less than 10% change in the area of the animal between consecutive samples

**For acute DAMGO administration experiments in the OF.** Different groups of C57BL/6 J mice received saline or DAMGO i.p. and were placed on the edge of the open field 30 min later. Mice were allowed to explore the arena for a total of 10 min.

**For acute stress experiments in the OF.** C57BL/6 J mice were placed in a CD1 mouse's home cage and left there for a total of 30 min. When an accumulated 20 sec of fight were completed, a transparent splitter was placed between the resident CD1 and the intruder test mouse to prevent further physical contact. Control mice were housed with a novel C57BL/6 J mouse for 30 minutes but no physical interaction was allowed. Then, mice were placed on the edge of the open field and were allowed to explore freely for a total of 10 min.

**For chemogenetic activation experiments in the OF.** TRAP2 mice were injected with c21 (1 mg/kg) and were placed in the open field 30 min later for a total duration of 10 min, as described above. Mice in the control group received a saline injection 30 prior to being placed in the open field arena.

**Binge-like food reward intake.** Palatable food choice experiments took place in a clean, enrichment-free cage with two metal food plates attached vertically to both opposing ends of the cage wall. One plate was mounted with regular chow (3.61 kcal/g) and the other plate was mounted with fat (Blanc de Boeuf, Vandermoortele, 9.0 kcal/g). Before starting the experiment, animals were habituated to the new fat diet

for 1 h per day until a baseline rate of fat intake was reached. Intake was measured by weighing the fat before and after the 1 h session. A steady baseline was acknowledged only when the average weight of fat consumed deviated no more than 0.05 g between subsequent days.

To gauge the effects of acute DAMGO administration on food reward intake, C57BL/6 J mice were injected daily with saline 15 min before the 1 h feeding session. When a steady baseline was reached, mice were injected with saline (control) or DAMGO (1 mg/kg) 15 min before the 1 h feeding session.

To gauge the effects of acute social stress, C57BL/6 J mice were placed in the feeding cage daily for 1 h. When a steady baseline was reached, mice were co-housed with a novel male C57BL/6 J conspecific (control) or were exposed to an aggressor CD-1 mouse 15 min before the 1 h feeding session.

To determine the effects of chemogenetic activation/inhibition of VTA$_{DAMGO}$ or VTA$_{STRESS}$, TRAP2 mice were placed daily in the feeding cage as described above. When a steady baseline was reached, mice were injected with saline or c21 (hM3Dq: 1 mg/kg, hM4Di: 2 mg/kg) 30 min before the start of the 1h-long feeding session. After a washout period (minimum 5 days), the experiment was repeated with mice receiving saline or c21 in a counter-balanced manner.

**For food intake necessity experiments in VTA ensemble mCherry or hM4Di-expressing mice.** For DAMGO-induced hyperphagia, on the test day mice, in which previously DAMGO-TRAPed ensembles were targeted to express hM4Di, received c21 (2 mg/kg) i.p. 15 min before injection of DAMGO and were placed in the feeding cage 15 min later. During baseline and post-test days, mice received a saline injection 15 min prior to being placed in the feeding cage for a total of 1 hour. For acute stress-induced hypophagia on the test day, half of the mice received a c21 injection (2 mg/kg) and the other half received a saline injection 15 min before being exposed to the social stress paradigm. 15 min after the stress episode mice were placed in the feeding cage for 1 hour, where after the chow and fat intake was determined. A minimum of 5 days later, the experiment was repeated in a counter-balanced manner as described above.

**Optogenetic intracranial self-stimulation.** A minimum of 10 days following TRAPing of VTA$_{DAMGO}$ or VTA$_{STRESS}$, mice were placed daily in an operant chamber (30.5 cm × 24.1 cm × 33 cm, MedPC Associates), featuring a curved rear wall and a grid floor. Five cue ports (2.54 cm × 2.2 cm × 2.54 cm), each equipped with an infrared beam, were integrated into the curved wall. A single cue port was assigned as the "active" port. A nose-poke in the active port resulted in 2 s of 473-nm light (10 ms, 20 Hz, -8 mW) delivered via a diode-pumped solid state (DPSS) laser (Changchun New Industries Optoelectronics) that was controlled by MedPc software. Nose-poking at inactive ports was recorded but had not programmed consequences. For laser shutoff experiments, mice were placed into the operant apparatus where they received continuous 473-nm laser stimulation (10 ms, 20 Hz -8 mW, interstimulus interval: 2 s). Nose-poking in the active port resulted in laser shutoff for 20 s. Responses in the inactive ports were recorded but had no programmed consequences. Intracranial self-stimulation (ICSS) and laser shutoff experiments were performed in two batches in a counter-balanced manner. To confirm that mice will nose-poke specifically for 473-nm light stimulation (or laser shutoff), the same experiments were performed nose-poking was able to control green (532-nm) laser stimulation. Mice performed each task for 3 days in a counter-balanced manner with a minimum washout period of 2 days in between.

**Fiber photometry**
Fiber photometry experiments were performed using a Doric setup, which included blue (465 nm) and purple (405 nm) light-emitting diodes. These light sources were connected to a high numerical

aperture (0.48), large core (400 μm) optical fiber patch cord, which was attached to a matching brain implant in each mouse. GCaMP8s fluorescence was collected by the same fiber, split by two dichromatic mirrors (420-450 nm and 460–490 nm), and transmitted through 600 μm core cables to photoreceivers (Newport 2151 with lensed FC adapter). The Doric software managed the LEDs and independently demodulated fluorescence brightness resulting from 405 nm and 465 nm excitation. Data acquisition was performed at 12 kilosamples per second and downsampled by a factor of 50 using the Doric Neuroscience Studio software. The software also calculated ΔF/F0 as 100* (F-F0)/F0, where F0 was the running average over a 60-second time window. Custom Python scripts were used to synchronize the onset of each behavior with the photometric ΔF/F$_0$ signal. For statistical comparisons, the average ΔF/F$_0$ was calculated per animal (of one hemisphere with optimal fiber placement) using the following time bins: baseline (−2.25 to −1 seconds prior to TTL pulse) and onset (0 to 1.25 seconds after the TTL pulse).

**Fat intake.** Experiments were conducted in an operant chamber (30.5 cm × 24.1 cm × 33 cm, MedPC Associates) with a curved rear wall and a grid floor. The curved wall featured five cue ports (2.54 cm × 2.2 cm × 2.54 cm), each equipped with an infrared beam. One cue port contained a piece of beef fat (Blanc de Boeuf, Vandermoortele, 9.0 kcal/g). Mice were acclimated to the chamber for 2 hours with unrestricted access to the fat (placed in a port) before photometric measurements. Nose pokes in the port with fat, registered by MedPC software through infrared beam interruptions, served as an indicator of fat access. The Doric setup produced and recorded a TTL pulse for each nose poke, with calcium transients synchronized to these pulses and averaged per animal.

**Acute foot shocks.** Mice were placed in a modified operant chamber (29.53 × 24.84 × 18.67 cm, Med Associates) designed to administer foot shocks, and given a 5-minute acclimation period. A total of 20 foot shocks, each lasting 1 second with an intensity of 0.7 mA, were applied at 90-second intervals. Calcium transients were measured and synchronized to the onset of each shock, then averaged for each individual subject.

### Immunohistochemistry and microscopy
Animals were transcardially perfused with 4% paraformaldehyde (PFA) and brains were post-fixated in 4% PFA at 4 °C for 24 -48h. Next, they were transferred to a 30% sucrose solution in PBS where they remained for another 48–72 h. Coronal sections of 35 μm were made with a cryostat (CM1950, Leica, Netherlands). Slices were first washed 4 × 10 min in PBS followed by incubation in blocking solution [5% normal goat serum (NGS), 2.5% bovine serum albumin, 0.2% Triton X-100 in PBS] for 1 h. Subsequently, slices were moved to the primary antibody solution containing one or more of the following antibodies in blocking solution and were left there overnight: rabbit-anti-c-Fos (1:1000; Cell Signaling Technology, 2250 s, Leiden, Netherlands), chicken-anti-GFP (1:1000, Aves), rabbit-anti-RFP (1:1000, Rockland), mouse-anti-TH (1:1000; Sigma-Aldrich Chemie N.V, MAB318, Zwijndrecht, Netherlands), mouse-anti-NeuN (1:1000; Abcam, ab104224, Waltham, MA, United States). The following day slices were washed 4 × 10 min followed by 2 h incubation in secondary antibody solution containing goat-anti-rabbit Alexa Fluor 488 (1:500; Abcam, ab150077, Waltham, MA, United States), goat-anti-chicken Alexa Fluor 488 (1:500; Abcam, ab150169, Waltham, MA, United States) and goat-anti-mouse Alexa Fluor 647 (1:500; Thermo Fischer Scientific, A-21236, Waltham, MA, United States). Slices were then moved to DAPI (0.5 μg/ml; Biotium brand, VWR, #89139-054, Netherlands) for 5 min followed by 2 × 10 min washes in PBS. Finally, slices were mounted on glass slides using 0.2% gelatin in PBS and coverslipped using FluorSave (Millipore, 345789, Amsterdam, Netherlands).

For cell counting in the VTA, 5-6 images of a VTA hemisection were obtained using a wide-field epifluorescence microscope (AX10, Zeiss). Specifically, a multi-channel image was captured, utilizing tyrosine hydroxylase (TH) staining to establish the VTA's boundaries. Fiji software[53] was employed to manually delineate the boundaries of VTA hemisections, using the TH staining outlines. Subsequently, an experimenter, unaware of the experimental conditions, manually counted cells expressing Fos or tdTomato. Cell density was then calculated as the number of Fos$^+$ or tdTomato$^+$ cells relative to the area of the designated region of interest. For co-localization analysis 4−6 z-stacks were acquired using a confocal microscope (LSM 880, Zeiss). For topographical analysis of Fos$^+$ cells, the X, Y coordinates of Fos$^+$ were extracted and their distance from a reference point (Suppl. Figure 2F) was calculated. Then the average X and Y coordinate was calculated separately for rostral (−2.92 < −3.28 mm relative to bregma) middle (−3.28 < −3.52 mm relative to bregma) and caudal (−3.52 < −3.80 mm relative to bregma) VTA slices.

### Data analysis and statistics
Data were analyzed using GraphPad Prism 9.3.1 (San Diego, CA, United States), Noldus (Ethovision V11.0), Python v3.8.8, MedPC-4 (Med Associates Inc), and MATLAB (Mathworks, Natick, MA, United States). Animals were randomly assigned to experimental conditions. Data are always reported and represented as mean + SEM, with single data points plotted representing individual animals. No animals were excluded unless otherwise specified. Normality was established using the Kolmogorov-Smirnov test. Where appropriate, statistical tests employed included One-way ANOVAs, Two-way ANOVAs, or Multi-way Repeated-measures ANOVAs. In instances where the omnibus ANOVAs indicated significant results, post hoc contrasts were conducted to further analyze the data. Two-tailed test were always performed with α = 0.05.

### Reporting summary
Further information on research design is available in the Nature Portfolio Reporting Summary linked to this article.

## Data availability
All detailed outcomes of statistical test are provided in Supplementary data. Source data are provided with this paper. The datasets generated during and/or analyzed during the current study are available from the corresponding author on request.

## Code availability
Codes used during the current study are available from the corresponding author on request.

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

## Acknowledgements

We thank Danai Riga, Rogier Poorthuis, and the entire Meye Lab for discussions and critical reading of the manuscript. We thank Nicky van Kronenburg and Laura Supiot for assistance in mouse breedings. This work was supported by (to FM) the NWO Gravitation project BRAIN-SCAPES: A Roadmap from Neurogenetics to Neurobiology (024.004.012), the ERC under the European Union's Horizon 2020 research and innovation program (grant agreement 804089; ReCoDE), and the NWO VIDI grant 203.102.

## Author contributions

I.K. performed and analyzed all histological experiments and analyzes, supported by Svd.S. and D.D. I.K., I.W.-D. and M.L. performed stereotactic surgeries. I.K. performed and analyzed all behavioral experiments and analyses, supported by L.P. F.M., designed the study with I.K. F.M. and I.K. wrote the manuscript with the help of RAHA and all other authors.

## Competing interests

The authors declare no competing interests.
