## [Transparent Peer Review file · Nature Communications]

Distinct ventral tegmental area neuronal ensembles are indispensable for reward-driven approach and stress-driven avoidance behaviors

Corresponding Author: Dr Frank Meye

Version 0:

Reviewer comments:

Reviewer #1

(Remarks to the Author)

In this manuscript by Koutlas et al, the authors tackle the intriguing question of how valence is encoded by heterogeneous ensembles of VTA neurons, including those expressing dopamine, GABA, and glutamate. They report largely non-overlapping groups of neurons (of all neurotransmitter types) express Fos after systemic DAMGO and acute social stress, respectively. These ensembles are reactivated by stimuli of similar valence (fat consumption and foot shock) and most of the behavioral sequelae of DAMGO vs social stress can be recapitulated or blocked by respective chemogenetic activation or blockade. The question is novel, the paper is well written, and I appreciate the multiple approaches used to address the experimental hypotheses. I think this paper will be of wide interest to neuroscience community, but I have a few questions and comments that need to be addressed before publication.

Major

1. As the authors note, previous studies have reported dopamine neurons that respond either to stress or reward. Since 50% of the ensemble are dopamine neurons, can the results entirely be explained by these distinct dopamine populations? Could the results be recapitulated if restricted to dopamine, glutamatergic, or GABAergic neurons? A key experiment needed to support their claim that ensembles of neurons with different neurotransmitters encode different valence is to show whether one cell type or the entire ensemble drive these effects. The ideal way would be to use an intersectional approach between the TRAP and DAT/VGAT/VGLUT-cre lines to express DREADD selectively in tagged neurons of a specific neurotransmitter. However, if this proves too experimentally challenging, an alternative would be to counter the effect of TRAP with global VTA DAT/VGAT/VGLUT-cre manipulations. For example, is VTADAMGO ensemble hM3 activation prevented by VTA DAT-cre hM4 inhibition?

2. Figure 2 appears to be missing some data valuable for the interpretation of the results. For example, the authors show about 15% overlap in Fos+ cells after DAMGO/stress or stress/DAMGO (Figure 2H). How much overlap would result from naïve/naïve, naïve/DAMGO, or naïve/stress?

3. The discussion could be enriched by addressing these questions:

A. What are the interactions within an ensemble of neurons with neurotransmitter types? Obviously, fiber photometry does not allow this to be addressed.

B. How do prior findings of salience + reward (see <https://www.ncbi.nlm.nih.gov/pmc/articles/PMC4826767/>) encoded in the same neurons fit with c-fos suggesting only a subset of neurons encode salience? Moreover, wouldn't dopamine neurons that fire both to rewarding stimuli and the relief from stress (doi: 10.1016/j.neuron.2023.09.004. PubMed PMID: 37776853) be included in both ensembles?

C. How do the authors reconcile their results with previous studies showing topographic organization (e.g., <https://doi.org/10.1016/j.neuropharm.2013.03.019> and [10.1038/s41593-021-00898-2](https://doi.org/10.1038/s41593-021-00898-2))? Would the authors expect to see similar results if the fiber was located elsewhere?

Minor

1. The authors switch the way to present the data between different figures. For example, in figure 4D the CPP results are presented as post-pre, while in fig 5 E and F CPP results are presented as times in compartments. The authors should choose one way of data presentation and stick to it throughout the figures. Also, in fig 1, stressed mice moved significantly

less than their unstressed controls. Therefore, if we compare the results as a ratio of time spent in the center vs periphery for each group, I am not sure if results obtained in stressed group can be interpreted as elevated anxiety.

2. The authors use 400um optical fibers for fiber photometry experiments and 200um fibers for their optogenetic experiments. Fibers of both sizes should leave a clear track visible as a hole in the tissue where the fibers were implanted. However, in Fig 3 B and 4M the representative pictures lack fiber tracks, and the white track marked seems to not reflect the real fiber placement (no fiber track is visible in the top panel of fig 4M). The traces overlaid by the authors should reflect the actual fiber placements.

3. The abbreviation Cf is used to refer to a different, but non-contradictory, opinion. Use v. (viz) instead to refer the reader to a specific figure.

4. Line 385: of instead for

5. Lines 731 and 735 appear to be redundant.

6. 415: typo: hyphagic

Reviewer #2

(Remarks to the Author)

In this manuscript Koutlas et al. show that negatively and positively valenced stimuli recruit the activity of largely separate populations of ventral tegmental area (VTA) neurons, not defined based on neurochemical divisions. This is a rigorous sets of experiments using a complementary set of approaches to show that these ensembles are also activated by alternative stimuli of the same, but not different, valence, that they are sufficient to drive relevant conditioned behavior, and are required for the development of these behaviors. The analyses are appropriate and rigorous, including analyses based on individual animal averages for fiber photometry. I just have some minor comments.

1. While the analyses seem appropriate, there are some cases where the specific effects that are being reported in the figure legends are not clear. For instance, in the Figure 4 legend, a Two-way RM ANOVA is reported for the preference scores in 4D, which I think is reporting on the interaction, but it could be labelled more clearly. In some cases, it is also not clear whether direct comparisons have been made between mCherry and each experimental condition. For instance, is the post-pre score for mCherry in 4D significantly different from the post-pre score for the VTA-stress group?

2. The authors should indicate that subjects are male mice in the abstract

Reviewer #3

(Remarks to the Author)

This manuscript from Koutlas et al. identifies two neuronal ensembles within the VTA: one for reward (linked to opioid stimuli) and one for stress (linked to an acute social stress task). When mice are exposed to congruent valence stimuli, these populations of cells seem to preferentially reactivate, and their reactivation is essential for valence-related behavioral outcomes.

While I found this paper scientifically interesting, novel, and well-conducted, I have a few points I would like the authors to address regarding the main findings and their interpretation:

(1) Alternative Interpretation: All their results can be explained with a different interpretation that does not require their valence belonging: these two populations are processing two different events, which is why they are not overlapping. To prove that opposite valence determines the recruitment of different populations of cells in the VTA, and not merely the occurrence of two different events, one main experiment should be conducted: cFos/TRAP overlap, but checking DAMGO (or STRESS) population with another task (not the same) with the same valence. While it may seem that a similar experiment has been performed (see Figure 3, using fiber photometry to visualize the activity of VTA DAMGO or VTA STRESS populations in tasks with the same or opposite valence), these data are not fully convincing since VTA DAMGO reactivation levels are similar (see point below (4)) during food reward and foot-shock experiences. Moreover, it is still impossible to deduce from these results if there is a stronger overlap between events with the same valence. If DAMGO and STRESS are activating cells linked with their positive or negative valence, the overlap ratio will match this prediction. Otherwise, all interpretations of results should be revised to tone down the valence association of these cells.

(2) Abstract Clarity: The abstract could be reformulated to clarify the message. I found it confusing to refer to "segregated intermingled ensembles." Are they overlapping? In what sense are they intermingled? How do the results support this? Moreover, it is difficult to understand what "small" population means and how one could define "small" within a region. Small compared to what? Depending on point (1), interpretation may need a revision.

(3) Reporting Precision: There are other points in the manuscript where the way results are reported seems imprecise. For example:

Page 14, line 334: "...ensembles are to a considerable extent non-overlapping subsets." It is not clear what the authors mean by "considerable extent" and how we should interpret this statistically.

Line 340: "two ensembles have selectivity in responding to either a congruently positive or negative experience" is not based on results presented (see point (1)).

Page 9, line 197: "...sufficiently drives..." what does "sufficiently" mean?

Page 9, line 189: what is "salient"?

Page 15, line 348: "not overlapping for the most part" what does this mean?

(4) Statistical Comparisons: Statistical comparisons between VTA DAMGO responding to food and to shocks should be presented. In Figure 3, the range of $\Delta F/F$ is different between food and shocks, making the comparison difficult. If the question here is whether the VTA DAMGO population is preferentially engaged in responding to positive valence (and not negative), I believe this result fails to address the question. However, the way it is presented does not allow comparison between the two. If statistically compared, will VTA DAMGO be more strongly reactivated during food? This is the result I have the most problems understanding and giving importance to. It seems that VTA DAMGO are driven by both reward and shock, and if the hypothesis is that it should be more driven by reward, this result is not convincing because the fluorescence is identical for the two stimuli. Figure 3C: use the same range of $\Delta F/F$. To really understand this result, I propose again the experiment mentioned in point (1) which will better answer the question: VTA + and - valence recruits non-overlapping populations of cells while + valences (or - valences) recruit a more similar population. Note, using fiber photometry, we cannot discern if the effect of $\Delta F/F$ is driven by the entire population or a relatively minor part of it. With TRAP/Fos overlap we may be able to answer this.

(5) Optogenetic experiments: Authors should explain why they didn't use optogenetics for all modulation experiments. Is the behavior correlating with the quantity of virus expressed in the different animals? This is relevant considering the conclusions of small ensembles recruited. However, the number of cells infected and tagged is not shown.

(6) Figure 4 Analysis: Figures 4F, G, H: statistically comparing mCherry between the different graphs and using one graph for all may help understand possible differences between groups. Moreover, it is not clear if the same animals underwent two different tasks with vehicle and then C21. Methods seem to suggest so, but figure legends and stats don't. Please clarify and add a protocol in the figure to help understanding. If they are not the same animals, why?

Minor Points:

- Figure 2E: It is not clear what the significance refers to. Specifically, I would like to see comparisons between DAMGO and Stress subpopulations, and if the stars (*) are showing this, they should be plotted in the same graph to clearly visually show these differences.

- Figure Protocols: For all figure protocols, I suggest making it clearer when referring to the virus by writing "viral injection" or similar. Moreover, the day count does not match the methods section when the behavior always starts at day 1 and not after 30 days. I suggest matching the figure protocols with the methods, maybe by using -30 for the viral infections and day 1 for behavior in the figures. Use consistent terminology between figures and methods. Align with methods on page 24, starting line 579. Then add more details about the treatments in figures protocols, like for example Fig 4A when C21 is injected 30 minutes before.

- Figure 3B: Why didn't the authors use TH staining for better VTA identification? As with all staining images, I highly suggest increasing the contrast since when figures are printed, increased contrast and clarity in all figures help. In printed papers, figures often look completely black, and no observations can be made.

- Figure 4E: It seems mice may exhibit freezing behavior. Can you show the percentage of freezing?

- Figure 4N: Are these the same mice? How long is the test? Why are the "nose pokes/h" values so high already on D1? Can you show how the behavior evolves during training?

- Page 22, line 532: What do you mean by "novel" if you then use "mate"?

- DREADDs Nomenclature: When hm4d or hm3d is used, add it to the figures instead of generic DREADDs.

Reviewer #4

(Remarks to the Author)

Version 1:

Reviewer comments:

Reviewer #1

(Remarks to the Author)

The authors have fully addressed all our comments. We congratulate them on a compelling and interesting manuscript!

two minor edits:

1. Figure 2 legend typo: add space between opposite valence in title
2. please substantially increase the font size of supplementary table 1

Reviewer #2

(Remarks to the Author)

The authors have addressed all my prior concerns and further strengthened the manuscript with valuable new experiments.

Reviewer #3

(Remarks to the Author)

The authors have thoroughly addressed all of my comments from the previous round of review. I have no remaining concerns and remain enthusiastic about the manuscript's novelty and significance.

Reviewer #4

(Remarks to the Author)

Dear Editor,

We are happy to see that the reviewers consider our study of interest and we are thankful for the comments they have provided to help further strengthen the manuscript. To address their comments we have performed multiple new experiments and analyses. Please find below the point-by-point discussion with regards to the reviewer comments, with our answers in blue. Also, the textual changes that we made in the manuscript in relation to reviewer comments have been made there in blue.

REVIEWER #1 (R1)

In this manuscript by Koutlas et al, the authors tackle the intriguing question of how valence is encoded by heterogenous ensembles of VTA neurons, including those expressing dopamine, GABA, and glutamate. They report largely non-overlapping groups of neurons (of all neurotransmitter types) express Fos after systemic DAMGO and acute social stress, respectively. These ensembles are reactivated by stimuli of similar valence (fat consumption and foot shock) and most of the behavioral sequelae of DAMGO vs social stress can be recapitulated or blocked by respective chemogenetic activation or blockade. The question is novel, the paper is well written, and I appreciate the multiple approaches used to address the experimental hypotheses. I think this paper will be of wide interest to neuroscience community, but I have a few questions and comments that need to be addressed before publication.

We thank the reviewer for their positive evaluation of our work.

Major:

1. As the authors note, previous studies have reported dopamine neurons that respond either to stress or reward. Since 50% of the ensemble are dopamine neurons, can the results entirely be explained by these distinct dopamine populations? Could the results be recapitulated if restricted to dopamine, glutamatergic, or GABAergic neurons? A key experiment needed to support their claim that ensembles of neurons with different neurotransmitters encode different valence is to show whether one cell type or the entire ensemble drive these effects. The ideal way would be to use an intersectional approach between the TRAP and DAT/VGAT/VGLUT-cre lines to express DREADD selectively in tagged neurons of a specific neurotransmitter. However, if this proves too experimentally challenging, an alternative would be to counter the effect of TRAP with global VTA DAT/VGAT/VGLUT-cre manipulations. For example, is VTADAMGO ensemble hM3 activation prevented by VTA DAT-cre hM4 inhibition?

The reviewer addresses an important point. We followed the suggestion of the reviewer on what the ideal manner would be to address this issue, and performed new experiments in which we targeted either the dopaminergic or the GABAergic subset of the VTA_{SOCIAL STRESS} or the VTA_{DAMO} ensembles. We did this by crossing the TRAP2 mice with other transgenic mice expressing FLP in GABAergic (VGAT-FLP) or

mainly dopaminergic (Pitx3-FLP) cells. We injected these mice with intersectional viruses to either optogenetically stimulate (AAV-CreON-FlpON-ChR2) or chemogenetically inhibit (AAV-CreON-FLPON-hM4Di) the specific subsets of the ensemble.

We then tested these mice both in intracranial self-stimulation paradigms, and in place preference/avoidance paradigms. These results are shown in new Figure 6 and Supplementary Figure 6 in the manuscript. In summary, we observed that:

Mice self-stimulated $VTA_{DAMGO-DOPAMINE}$ cells, and also self-stimulated $VTA_{DAMGO-GABA}$ cells (Fig 6A-E; Suppl. Fig 6A). Contrarily, mice actively turned off stimulation of either the $VTA_{STRESS-DOPAMINE}$ cells or of $VTA_{STRESS-GABA}$ cells (Fig 6A-F; Suppl. Fig 6B).

Furthermore, when we chemogenetically inhibited the activity of $VTA_{STRESS-DOPAMINE}$ or of $VTA_{STRESS-GABA}$ neurons during social stress, the mice no longer avoided the associated compartment in the place avoidance test. This suggests that activity of both these subsets within the VTA_{STRESS} ensemble is indispensable for the emergence of place avoidance (Fig. 6G-J). When we blocked the activity of $VTA_{DAMGO-DOPAMINE}$ neurons during DAMGO, this attenuated place preference for the DAMGO compartment. However, blocking the activity of $VTA_{DAMGO-GABA}$ neurons was not sufficient to prevent DAMGO-associated place preference (Fig. 6G-J).

Overall, these findings suggest that within the VTA_{DAMGO} and VTA_{STRESS} ensembles, the different neurotransmitter-defined populations play congruent roles in valence processes. Moreover, they suggest that at least for the encoding of social stress-driven avoidance, multiple NT-defined subsets of the VTA_{STRESS} ensemble are indispensable.

2. Figure 2 appears to be missing some data valuable for the interpretation of the results. For example, the authors show about 15% overlap in Fos+ cells after DAMGO/stress or stress/DAMGO (Figure 2H). How much overlap would result from naïve/naïve, naïve/DAMGO, or naïve/stress?

We have now performed a new experiment to address this question. We administered 4OH tamoxifen in naïve mice (home cage group), then provided another experience (home cage, DAMGO or social stress) and performed cFos immunohistochemistry afterwards, and evaluated for extent of overlap (Suppl. Fig. 2Q). We observed that there was no significant difference between on the one hand the extent of home cage/home cage, home cage/DAMGO and home cage/social stress overlap, and on the other hand the extent of overlap in DAMGO/social stress, or social stress/DAMGO conditions. These data suggest that the ~15% overlap that we observe in the DAMGO/social stress and social stress/DAMGO conditions cannot be readily interpreted as reflecting dual valence encoding. We now discuss the potential salience encoding of the VTA ensembles also in light of these new findings.

3. The discussion could be enriched by addressing these questions:

We thank the reviewer for their suggestions, and have added Discussion on their points (see below).

A. What are the interactions within an ensemble of neurons with neurotransmitter types? Obviously, fiber photometry does not allow this to be addressed.

We have now added Discussion on this point in relation to the results obtained from our new experiment (point 1 of the reviewer, New Figure 6) in which we show that within an ensemble, different neurotransmitter-defined neuronal types have functions in accordance with the ensemble function. For instance such that stimulation of the GABAergic and of dopaminergic neurons of the VTA_{DAMGO} ensemble is reinforcing, whereas this is not the case for either subtype of the VTA_{STRESS} ensemble (where stimulation of either is avoided). We discuss that while it is indeed known that there is at least a certain extent of local connection between VTA_{GABA} neurons and VTA_{DOPAMINE} neurons, our findings could suggest that the GABA neurons within a given ensemble may not inhibit the dopamine neurons of that same ensemble. We identify this as an interesting opportunity for future research, and this is indeed also something we are embarking on ourselves.

B. How do prior findings of salience + reward (see <https://www.ncbi.nlm.nih.gov/pmc/articles/PMC4826767/>) encoded in the same neurons fit with c-fos suggesting only a subset of neurons encode salience? Moreover, wouldn't dopamine neurons that fire both to rewarding stimuli and the relief from stress (doi: 10.1016/j.neuron.2023.09.004. PubMed PMID: 37776853) be included in both ensembles?

We have added further discussion on this point, also incorporating our new data on home cage controls (point 2 of the reviewer; Suppl. Fig. 2M-Q). In general we state that the current c-fos based techniques we employ in this study (as opposed to for instance electrophysiological techniques) may be less suitable to identify to which extent some of the ensemble also encode salience, and what the specific percentage of neurons within an ensemble is that does so. Indeed we found that also home cage controls (for which it should be noted that they do still receive salient i.p. injections) showed a basal level of overlap between TRAP and Fos staining upon (home cage) re-exposure that was not lower than what we observed in terms of overlap for social stress and DAMGO dual responsivity of neurons (~15%, Fig. 2H). It remains therefore possible that the ~15% of neurons showing overlap to DAMGO and Social Stress in Fig. 2H, do not necessarily reflect either salience. We elaborate on this in the Discussion, and state that salience coding is better addressed with in vivo electrophysiological measures.

Regarding whether our stress-TRAP approach shouldn't also capture subsequent 'relief' neurons. We think this occurrence may have been limited by the methodological choice we made regarding the labeling/tagging of the VTA_{SOCIAL STRESS} ensemble. After the accumulated fighting period (20s), mice were placed back in the co-housing situation with the aggressor (though separated by barrier). Meaning that immediately after the main aversive experience (fight), there was likely still a continuation of some level of aversive experience (separated co-housing with aggressor). This may have limited the amount of relief experienced in the immediate aftermath, and therefore the amount of recruitment of relief-encoding neurons in the VTA_{STRESS} ensemble. The cohousing part, which we had described in the methods, we have now also made more explicit in the figure schematics for clarity.

C. How do the authors reconcile their results with previous studies showing topographic organization (e.g., <https://doi.org/10.1016/j.neuropharm.2013.03.019> and [10.1038/s41593-021-00898-2](https://doi.org/10.1038/s41593-021-00898-2)? Would the authors expect to see similar results if the fiber was located elsewhere?

The reviewer is correct that in the VTA at large there is certainly evidence for a degree of topographical organization in at least VTA output projections and to some extent also behavioral functions. Though it is also the case that for behavioral roles these topographical organizations are not fully black-white. Our study deviates from the cited studies in that it takes an ensemble view, based on Fos, and we did not find direct evidence for topographic differences at the level of (DAMGO / social stress) ensembles. We are however mindful of limitations of c-fos based approaches as read-outs of neuronal activity. We therefore take our results to mean that VTA neurons with opposite roles in valences processes are at least in part intermingled, while leaving the possibility open that at a larger scale certain topographical trends do exist in the VTA, in terms of where it is relatively more likely to find neurons with certain (electrophysiological responses to) valence properties.

Minor

1. The authors switch the way to present the data between different figures. For example, in figure 4D the CPP results are presented as post-pre, while in fig 5 E and F CPP results are presented as times in compartments. The authors should choose one way of data presentation and stick to it throughout the figures.

We would like to clarify that the differences in data presentation between Figure 4 and Figure 5 (also Figure 1) are due to the distinct types of tests employed in these experiments. In Figures 1 and 5, we used a single exposure place preference (sePP) test, which does not include a pre-test session. Therefore, presenting the data as time spent in compartments during the post-test session is the only way to present these results. On the other hand, Figure 4 displays data from a different version of the test involving a pre-test session and three conditioning days. Here, presenting the results as post-pre (i.e., post-test minus pre-test) enables us to capture and present the shift in preference induced by the conditioning sessions within mice. In our opinion, presenting the data as time spent in compartments during the post-test session alone would in this case result in a loss of valuable information. We understand that this confusion may have in part arisen because of the absence of an explicit timeline schematic for Figure 4. Therefore we have now added this for clarity (Suppl. Fig. 4A). Moreover, in the Results section we distinguish more explicitly between single exposure and repeated pairings versions of the preference/avoidance tests for clarity.

Also, in fig 1, stressed mice moved significantly less than their unstressed controls. Therefore, if we compare the results as a ratio of time spent in the center vs periphery for each group, I am not sure if results obtained in stressed group can be interpreted as elevated anxiety.

To address this point we have now also run the analysis of time spent in the center in Figure 1G using distance moved as a covariate, to correct for its possible influence on the measurement of time spent in the center. Doing so still yields a similar significant effect of stress on time spent in

the center ($F_{1,21}=7.089$, $p=0.015$). This indicates that the effect of stress on time spent in the center, which we interpret to reflect anxiety, was not a direct consequence of alterations in locomotor activity. We have now added this analysis with a covariate to the legend of this panel.

2. The authors use 400um optical fibers for fiber photometry experiments and 200um fibers for their optogenetic experiments. Fibers of both sizes should leave a clear track visible as a hole in the tissue where the fibers were implanted. However, in Fig 3 B and 4M the representative pictures lack fiber tracks, and the white track marked seems to not reflect the real fiber placement (no fiber track is visible in the top panel of fig 4M). The traces overlaid by the authors should reflect the actual fiber placements.

We thank the reviewer for this comment. We have improved the histological images.

3. The abbreviation Cf is used to refer to a different, but non-contradictory, opinion. Use v. (viz) instead to refer the reader to a specific figure.

Corrected.

4. Line 385: of instead for

Corrected.

5. Lines 731 and 735 appear to be redundant.

Corrected.

6. 415: typo: hyphagic

Corrected.

REVIEWER #2 (R2)

In this manuscript Koutlas et al. show that negatively and positively valenced stimuli recruit the activity of largely separate populations of ventral tegmental area (VTA) neurons, not defined based on neurochemical divisions. This is a rigorous sets of experiments using a complementary set of approaches to show that these ensembles are also activated by alternative stimuli of the same, but not different, valence, that they are sufficient to drive relevant conditioned behavior, and are required for the development of these behaviors. The analyses are appropriate and rigorous, including analyses based on individual animal averages for fiber photometry. I just have some minor comments.

We thank the reviewer for their positive comments.

1. While the analyses seem appropriate, there are some cases where the specific effects that are being reported in the figure legends are not clear. For instance, in the Figure 4 legend, a Two-way RM ANOVA is reported for the preference scores in 4D, which I think is reporting on the interaction, but it could be labelled more clearly. In some cases, it is also not clear whether direct comparisons have been made between mCherry and each experimental condition. For instance, is the post-pre score for mCherry in 4D significantly different from the post-pre score for the VTA-stress group?

We thank the reviewer for bringing this lack of clarity to our attention. We have now precised the performed analysis in the legend of Figure 4D. We clarify that there is a significant interaction between test time point (pre vs post) and TRAP condition (mCherry, DAMGO, Stress). On the basis of this omnibus ANOVA interaction, we then performed post-hoc tests for the three distinct TRAP conditions (mCherry, DAMGO, and Stress) with regards to their pre vs post score. The outcomes of those post-hoc tests are shown graphically in Figure 4D.

2. The authors should indicate that subjects are male mice in the abstract

We have now made this addition.

REVIEWER #3 (R3)

This manuscript from Koutlas et al. identifies two neuronal ensembles within the VTA: one for reward (linked to opioid stimuli) and one for stress (linked to an acute social stress task). When mice are exposed to congruent valence stimuli, these populations of cells seem to preferentially reactivate, and their reactivation is essential for valence-related behavioral outcomes.

While I found this paper scientifically interesting, novel, and well-conducted, I have a few points I would like the authors to address regarding the main findings and their interpretation:

We thank the reviewer for their evaluation of our work.

(1) Alternative Interpretation: All their results can be explained with a different interpretation that does not require their valence belonging: these two populations are processing two different events, which is why they are not overlapping. To prove that opposite valence determines the recruitment of different populations of cells in the VTA, and not merely the occurrence of two different events, one main experiment should be conducted: cFos/TRAP overlap, but checking DAMGO (or STRESS) population with another task (not the same) with the same valence. While it may seem that a similar experiment has been performed (see Figure 3, using fiber photometry to visualize the activity of VTA DAMGO or VTA STRESS populations in tasks with the same or opposite valence), these data are not fully convincing since VTA DAMGO reactivation levels are similar (see point below (4)) during food reward and foot-shock experiences. Moreover, it is still impossible to deduce from these results if there is a stronger overlap between events with the same valence. If DAMGO and STRESS are activating cells linked with their positive or negative valence, the overlap ratio will match this prediction. Otherwise, all interpretations of results should be revised to tone down the valence association of these cells.

The reviewer is correct that this is an important matter to resolve. We have therefore followed their advice and performed a new experiment in which we assessed the extent of overlap between (TRAPed) VTA_{DAMGO} cells with another reward (food reward) as assessed by cFos staining, and TRAPed VTA_{SOCIAL STRESS} cells with another aversive event (foot shock) assessed by cFos staining. These results are reported in Supplementary Fig. 2M-P. In summary: we report a significantly greater overlap in neuronal labeling in the VTA between DAMGO/food reward, as compared to the degree of overlap with DAMGO / social stress. Moreover, there was a significantly greater overlap between social stress / foot shock than with social stress / DAMGO. Overall, these data suggest that VTA_{DAMGO} cells are preferentially activated by rewarding experiences (DAMGO or food reward) as compared to a stressor (social stress), and that VTA_{SOCIAL STRESS} cells are preferentially activated by aversive experiences (social stress or foot shock) as compared to a reward (DAMGO).

(2) Abstract Clarity: The abstract could be reformulated to clarify the message. I found it confusing to refer to "segregated intermingled ensembles." Are they overlapping? In what sense are they intermingled? How do the results support this? Moreover, it is difficult to understand what "small"

population means and how one could define “small” within a region. Small compared to what? Depending on point (1), interpretation may need a revision.

We have revised the abstract to avoid lack of clarity on these points.

(3) Reporting Precision: There are other points in the manuscript where the way results are reported seems imprecise. For example:

Page 14, line 334: “...ensembles are to a considerable extent non-overlapping subsets.” It is not clear what the authors mean by “considerable extent” and how we should interpret this statistically.

The basis for stating that there is considerable lack of overlap between the ensembles is that for non-congruent valence stimuli there is only limited overlap in activation of the ensembles (~15%), which is significantly less than is the case for congruent stimuli. We have now added further explanatory text on this in the discussion.

Line 340: “two ensembles have selectivity in responding to either a congruently positive or negative experience” is not based on results presented (see point (1)).

In view of the new experiment that we performed (see point 1), showing that the ensembles are also preferentially activated by different congruent valence stimuli than incongruent valence stimuli, we now have further evidence to support this statement (Suppl. Fig. 2N-P).

Page 9, line 197: “..sufficiently drives...” what does “sufficiently” mean?

Corrected (removed).

Page 9, line 189: what is “salient”?

Corrected (removed).

Page 15, line 348: “not overlapping for the most part” what does this mean?

Corrected (removed).

(4) Statistical Comparisons: Statistical comparisons between VTA DAMGO responding to food and to shocks should be presented. In Figure 3, the range of $\Delta F/F$ is different between food and shocks, making the comparison difficult. If the question here is whether the VTA DAMGO population is

preferentially engaged in responding to positive valence (and not negative), I believe this result fails to address the question. However, the way it is presented does not allow comparison between the two. If statistically compared, will VTA DAMGO be more strongly reactivated during food? This is the result I have the most problems understanding and giving importance to. It seems that VTA DAMGO are driven by both reward and shock, and if the hypothesis is that it should be more driven by reward, this result is not convincing because the fluorescence is identical for the two stimuli. Figure 3C: use the same range of $\Delta F/F$. To really understand this result, I propose again the experiment mentioned in point (1) which will better answer the question: VTA + and - valence recruits non-overlapping populations of cells while + valences (or - valences) recruit a more similar population. Note, using fiber photometry, we cannot discern if the effect of $\Delta F/F$ is driven by the entire population or a relatively minor part of it. With TRAP/Fos overlap we may be able to answer this.

We understand the concern of the reviewer. It is difficult to compare fiber photometric results between animals, due to inherent variability in exact fiber placement in relation to GCaMP-expressing cells. Therefore we agree with the reviewer that the better approach for this comparison is based on a TRAP/Fos overlap experiment. We therefore refer back to the new experiment that we performed and discussed in relation to the Reviewer's earlier point (point 1), which shows preferential activation of ensembles to congruent valence stimuli (Suppl. Fig. 2N-P).

(5) Optogenetic experiments: Authors should explain why they didn't use optogenetics for all modulation experiments. Is the behavior correlating with the quantity of virus expressed in the different animals? This is relevant considering the conclusions of small ensembles recruited. However, the number of cells infected and tagged is not shown.

Given that chemogenetic manipulations are a bit less invasive to the animal than optogenetics (no fiber placement), we opted for chemogenetics whenever we deemed that continuous stimulation (or inhibition) over minutes to hours was appropriate for the read-out in the behavioral task. Instead, when a higher temporal control over stimulation was essential (mainly for the intracranial self-stimulation experiments) we used optogenetics instead.

As requested by the reviewer, we have now added further information on the correlation of ensemble targeting and behavioral effects (Suppl. Fig. 4B). We did not observe a significant correlation in this regard. We interpret this to mean that within a certain range of targeting the ensemble there are similar behavioral effects.

(6) Figure 4 Analysis: Figures 4F, G, H: statistically comparing mCherry between the different graphs and using one graph for all may help understand possible differences between groups. Moreover, it is not clear if the same animals underwent two different tasks with vehicle and then C21. Methods seem to suggest so, but figure legends and stats don't. Please clarify and add a protocol in the figure to help understanding. If they are not the same animals, why?

For the open field tests described in these panels, the animals that received C21 were different mice from the ones that received vehicle. We opted for this strategy for two main reasons:

(1) to avoid having the same animal be exposed multiple times to the open field, which in our experience can make the mouse's performance in the second OF different from in the first one, adding variability and potential difficulties in interpretations.

(2) To ensure that the experimental groups have comparable histories. For instance, animals in the stress-hM3Dq group experienced acute stress during TRAPing and received c21 in the OF, potentially forming an aversive association with the arena, as supported by our CPP/CPA data. Instead, animals in the DAMGO-hM3Dq group underwent a rewarding experience and might form a rewarding association with the arena. Therefore we think that making comparisons when using animals with similar histories is more meaningful.

To further enhance clarity that the mice that received C21 were different from the mice receiving vehicle, we have now added a visual representation of the protocol to the figure as the reviewer suggested (Suppl. Fig. 4C). We have also revised the methods section for further clarity on this point.

Minor Points:

- Figure 2E: It is not clear what the significance refers to. Specifically, I would like to see comparisons between DAMGO and Stress subpopulations, and if the stars (*) are showing this, they should be plotted in the same graph to clearly visually show these differences.

We thank the reviewer for pointing out this lack of clarity. We have now changed the representation of this graph (with DAMGO and stress side-by-side), to make more clear which comparisons are made.

- Figure Protocols: For all figure protocols, I suggest making it clearer when referring to the virus by writing "viral injection" or similar. Moreover, the day count does not match the methods section when the behavior always starts at day 1 and not after 30 days. I suggest matching the figure protocols with the methods, maybe by using -30 for the viral infections and day 1 for behavior in the figures. Use consistent terminology between figures and methods. Align with methods on page 24, starting line 579. Then add more details about the treatments in figures protocols, like for example Fig 4A when C21 is injected 30 minutes before.

We thank the reviewer for their suggestion and we have made amendments in accordance in all of the figure protocols, and aligned them with methods.

- Figure 3B: Why didn't the authors use TH staining for better VTA identification? As with all staining images, I highly suggest increasing the contrast since when figures are printed, increased contrast and

clarity in all figures help. In printed papers, figures often look completely black, and no observations can be made.

We have now added an example of TH staining that we used to mark the contours of the VTA in the fiber photometry main figure (Suppl. Fig. 3A). We also thank the reviewer for their suggestion regarding enhancing contrast. For all the histological / microscopy images we have now increased the visibility.

- Figure 4E: It seems mice may exhibit freezing behavior. Can you show the percentage of freezing?

We now analyzed immobility scores and found them not to be affected by chemogenetic manipulations. We have now added immobility graphs for all hm3Dq groups (Suppl. Fig. 4D-F).

- Figure 4N: Are these the same mice? How long is the test? Why are the "nose pokes/h" values so high already on D1? Can you show how the behavior evolves during training?

These are indeed the same mice, and the test is one hour long. While we had stated these points in the methods, for extra emphasis we now also state it in the legend. Mice indeed exhibited immediately (without a prior training to Day 1) a high willingness to nose poke for the active port in these experiments. We agree that it is striking that when stimulating the entire ensemble the mice already learn to nose poke during the time course of the first session. Interestingly, we also performed another new experiment in which we use an intersectional viral strategy to stimulate the GABAergic or the dopaminergic cells within either the VTA_{DAMGO} or VTA_{STRESS} ensemble. When doing so we observe that mice also self-stimulate just the GABAergic or the dopaminergic part of the VTA_{DAMGO} ensemble, but not yet significantly so on day 1. Similarly, mice will also not yet terminate enforced stimulation of the GABAergic or dopaminergic part of the VTA_{STRESS} ensemble on the first day after the contingency switch (nose poke becoming turning a laser off rather than on). These new data are shown in Figure 6E-F.

- Page 22, line 532: What do you mean by "novel" if you then use "mate"?

We have now corrected this (they were novel B6 male conspecifics).

- DREADDs Nomenclature: When hm4d or hm3d is used, add it to the figures instead of generic DREADDs.

We have now corrected this.

Reviewer #4 (Remarks to the Author):

We thank R4 for their contribution.

Dear Editor,

We are pleased that all of the reviewers consider that we have fully addressed their comments. We thank them for their evaluation of the work and for their appreciative comments. We hereby address the last couple of comments.

REVIEWER #1 (R1)

The authors have fully addressed all our comments. We congratulate them on a compelling and interesting manuscript!

We thank the reviewer.

two minor edits:

1. Figure 2 legend typo: add space between opposite valence in title

Corrected.

2. please substantially increase the font size of supplementary table 1

Corrected.

REVIEWER #2 (R2)

The authors have addressed all my prior concerns and further strengthened the manuscript with valuable new experiments.

We thank the reviewer.

REVIEWER #3 (R3)

The authors have thoroughly addressed all of my comments from the previous round of review. I have no remaining concerns and remain enthusiastic about the manuscript's novelty and significance.

We thank the reviewer.

REVIEWER #4 (R4)

We thank the reviewer.

We remain at your disposal in case further information is required.

Sincerely,

Frank Meye, PhD